# Prospective de novo drug design with deep interactome learning

Kenneth Atz [1], Leandro Cotos[1], Clemens Isert [1], Maria Håkansson[2], Dorota Focht[2], Mattis Hilleke [1], David F. Nippa [3,4], Michael Iff[1], Jann Ledergerber [1], Carl C. G. Schiebroek [1], Valentina Romeo[3], Jan A. Hiss [1], Daniel Merk [4], Petra Schneider [1], Bernd Kuhn[3], Uwe Grether [3] & Gisbert Schneider [1] ✉

De novo drug design aims to generate molecules from scratch that possess specific chemical and pharmacological properties. We present a computational approach utilizing interactome-based deep learning for ligand- and structure-based generation of drug-like molecules. This method capitalizes on the unique strengths of both graph neural networks and chemical language models, offering an alternative to the need for application-specific reinforcement, transfer, or few-shot learning. It enables the "zero-shot" construction of compound libraries tailored to possess specific bioactivity, synthesizability, and structural novelty. In order to proactively evaluate the deep interactome learning framework for protein structure-based drug design, potential new ligands targeting the binding site of the human peroxisome proliferator-activated receptor (PPAR) subtype gamma are generated. The top-ranking designs are chemically synthesized and computationally, biophysically, and biochemically characterized. Potent PPAR partial agonists are identified, demonstrating favorable activity and the desired selectivity profiles for both nuclear receptors and off-target interactions. Crystal structure determination of the ligand-receptor complex confirms the anticipated binding mode. This successful outcome positively advocates interactome-based de novo design for application in bioorganic and medicinal chemistry, enabling the creation of innovative bioactive molecules.

Computational de novo design encompasses the autonomous generation of new molecules with desired properties from scratch[1,2]. Chemical language models (CLMs) are machine learning techniques designed to process and learn from molecular structures represented as sequences (e.g., simplified molecular input line entry system (SMILES)-strings[3]). CLMs have found numerous applications for the de novo design of novel bioactive molecules[4,5]. Transfer learning, also known as fine-tuning, is one of the most prevalent applications of CLMs in the field of molecular design[6–8]. Transfer learning in the context of CLMs can be conceptualized as a two-step process. In the first step, the CLM undergoes pre-training using a vast data set of bioactive molecules that is not specifically tailored for the task at hand. This initial phase focuses on developing a foundational understanding of chemistry and acquiring knowledge about the characteristics of drug-

[1]ETH Zurich, Department of Chemistry and Applied Biosciences, Vladimir-Prelog-Weg 4, 8093 Zurich, Switzerland. [2]SARomics Biostructures AB, Medicon Village, SE-223 81 Lund, Sweden. [3]Roche Pharma Research and Early Development (pRED), Roche Innovation Center Basel, F. Hoffmann-La Roche Ltd., Grenzacherstrasse 124, CH-4070 Basel, Switzerland. [4]Department of Pharmacy, Ludwig-Maximilians-Universität München, Butenandtstrasse 5, 81377 Munich, Germany. ✉e-mail: gisbert@ethz.ch

like chemical space[9]. In the second step, the pre-trained CLM is fine-tuned using a smaller data set comprising molecules that specifically represent the desired activity and property profile[10]. This process refines the CLM's ability to generate molecules with the desired characteristics. Once trained, the CLM can generate virtual molecular libraries tailored to the specific task at hand[11]. Some CLM approaches integrate reinforcement learning techniques, enabling an additional level of fine-tuning to optimize the properties of the generated molecules[12,13].

However, the utilization of transfer learning and reinforcement learning in CLMs entails additional machine learning steps, which can pose challenges in terms of speed and seamless integration within the design-make-test-analysis cycle in medicinal chemistry[14–16]. Furthermore, transfer learning can be particularly challenging when applied to a single fine-tuning molecule[7,17]. It may also present difficulties in structure-based design applications that rely on explicit information about the protein binding site[18–21]. Although various structure-based de novo design methods have been introduced, their prospective applications have not been extensively explored, highlighting the need to fully assess the potential of these methods in practical scenarios[22,23].

Recent advancements have focused on studying molecular interaction networks, known as interactomes, which encompass various types of interactions such as protein-protein interactions, drug-target interactions, and drug-drug relationships. Analyzing these interactomes enables the prediction of previously unknown interactions and provides insights into the network topology[24–27]. Studying molecular interaction networks as a holistic entity offers a distinct advantage by allowing the analysis of long-range relationships between different nodes that are connected through multiple edges. This approach enables a comprehensive examination of the interconnectedness and dependencies among various components within the network[24].

To address the goal of studying the drug-target interactome comprehensively, we propose an approach that combines a CLM with interactome-based deep learning (Fig. 1a, b). This approach incorporates a neural network architecture consisting of a graph transformer neural network (GTNN) and a CLM utilizing a long-short-term memory (LSTM) (Fig. 1c, d, e). Herein, the deep learning model resulting from this approach is named DRAGONFLY (Drug-target interActome-based GeneratiON oF noveL biologicallY active molecules). Unlike conventional CLMs that rely on transfer learning with individual molecules, the method leverages interactome-based deep learning, which enables the incorporation of information from both, targets and ligands across multiple nodes. DRAGONFLY is capable of processing small-molecule ligand templates as well as three-dimensional (3D) protein binding site information. It operates on diverse chemical alphabets and does not require fine-tuning through transfer or reinforcement learning specific to a particular application. Furthermore, it enables the incorporation of desired physical and chemical properties into the generation of output molecules. This study introduces the prospective application of DRAGONFLY to structure-based de novo design, specifically for the generation of ligands with desired bioactivity profiles addressing one or multiple specific macromolecular targets (Fig. 1f).

## Results
### DRAGONFLY enables ligand- and structure-based molecular design
The central component of DRAGONFLY is its drug-target interactome, which captures the connections between small-molecule ligands and their macromolecular targets. This interaction can be depicted as a graph, where nodes represent bioactive ligands and their corresponding macromolecular targets (Fig. 1a). Distinct nodes were used to differentiate between orthosteric and allosteric binding sites within the same target. Edges were established between ligands and proteins that have an annotated binding affinity of less than or equal to 200 nM

(Fig. 1a) (values extracted from the ChEMBL database[28]). As a result of this procedure, an interactome was generated that consisted of ~360,000 ligands, 2989 targets, and around 500,000 bioactivities. This interactome was specifically designed for ligand-based design applications. In the case of structure-based design, only macromolecular targets with known 3D structures were considered, resulting in an interactome containing around 208,000 ligands, 726 targets, and around 263,000 bioactivities. This data structure based on the interactome facilitated the training of two deep learning models, specifically for ligand-based and structure-based de novo design (Fig. 1b).

The neural networks employed in the study accept a molecular graph as their input signal. In particular, a 3D graph was utilized for binding sites, while a 2D molecular graph was used for ligands (Fig. 1c and d). Subsequently, the input graph undergoes a transformation into SMILES-strings, which represent molecules with the desired bioactivity and physicochemical properties. This translation process was achieved by utilizing a graph-to-sequence deep learning model that combines a graph transformer neural network[29–31] with a long-short term memory (LSTM) neural network[32] (Fig. 1e). The selection of the graph-to-sequence architecture was made to facilitate the development of deep learning models capable of supporting both ligand-based and structure-based molecular design.

### DRAGONFLY considers synthesizability, novelty, bioactivity, and physicochemical properties for ligand design
The theoretical evaluation of DRAGONFLY focused on investigating the incorporation of specific physical and chemical properties into the DRAGONFLY model, as depicted in Fig. 2a. This evaluation revealed Pearson correlation coefficients ($r$) greater than or equal to 0.95 for all assessed physical and chemical properties. These properties included molecular weight ($r = 0.99$), rotatable bonds ($r = 0.98$), hydrogen bond acceptors ($r = 0.97$), hydrogen bond donors ($r = 0.96$), polar surface area ($r = 0.96$), and lipophilicity expressed as MolLog$P$[33] ($r = 0.97$). These high correlation coefficients indicate a strong relationship between the desired properties and the actual properties of the generated molecules.

The study also included the evaluation of novelty, synthesizability, and predicted bioactivity for the generated molecules. These criteria were essential in assessing the practicality and potential value of the designed compounds. To quantify molecular novelty, a rule-based algorithm was utilized, which captured both, scaffold and structural novelty. This algorithm, described in Equations (9)–(12), offers a quantitative measure of the uniqueness of each molecule in terms of its chemical structure. Synthesizability was assessed using the retrosynthetic accessibility score (RAScore), a recently published metric that assesses the feasibility of synthesizing a given molecule[34].

Additionally, to estimate the on-target bioactivity of the de novo designs, quantitative structure-activity relationship (QSAR) models were developed. The models utilized kernel ridge regression (KRR)-based machine learning[35], and were trained on three molecular descriptors: ECFP4[36], unscaled CATS[37], and USRCAT[38]. The descriptors used in the study encompassed a wide range of structural, pharmacophore, and shape-based similarities of the molecules, offering a comprehensive representation of their characteristics. A combination of descriptors, including ECFP for structural features as well as CATS and USRCAT for "fuzzy" features, was employed to capture both, specific and general molecular attributes. By incorporating these descriptors, the study aimed to facilitate the identification of molecular similarities between the highly ranked de novo designs and known bioactive compounds. For the majority of the 1265 investigated targets, the mean absolute errors (MAEs) for the predicted pIC$_{50}$ values were equal to or less than 0.6 (Fig. 2b, Fig. S2). Moreover, the KRR models have shown superior performance to decision tree baseline methods including gradient boosting and extreme gradient boosting

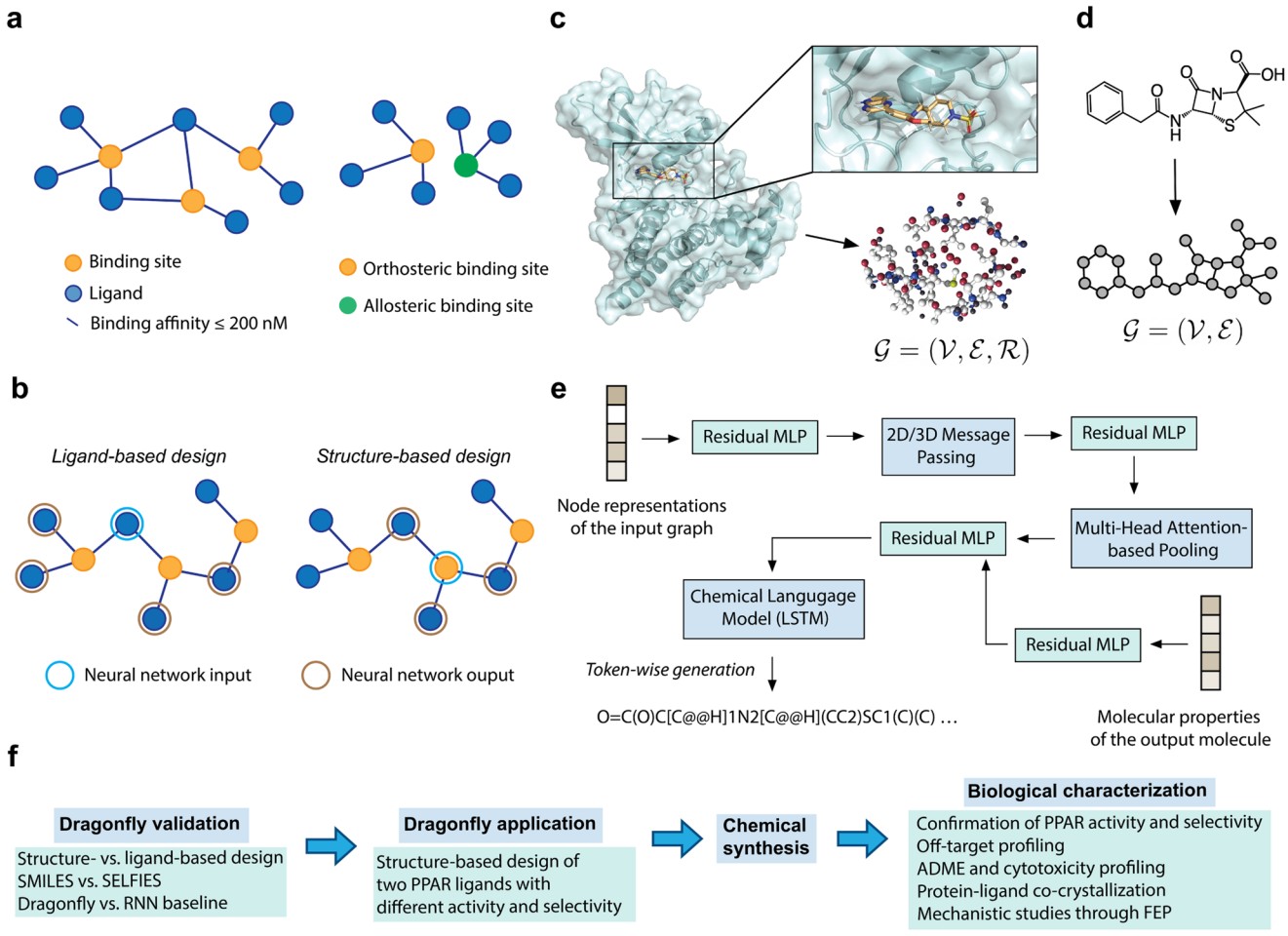

**Fig. 1 | DRAGONFLY architecture and workflow. a** Left: To construct the drug-target interactome graph, the targets are connected to their corresponding ligands based on reported bioactivities in the ChEMBL database[28]. Specifically, a connection is established between a ligand (blue circle) and its corresponding target (orange circle) if the ligand has been reported with a bioactivity equal to or <200 nM. Right: By representing allosteric and orthosteric binding sites as separate nodes (shown in green and orange, respectively), the drug-target interactome graph captures the specific interactions and relationships associated with each type of binding site. **b** Left: During the training phase for ligand-based design, a ligand molecule (represented as a blue circle) is taken as the input to the model. The desired output molecules (represented as brown circles) are selected based on their connection to the input molecule through a common node, indicating that they share a binding site. Right: For structure-based design, the input for the model is the binding site itself, represented as a blue circle. The desired output molecules, represented as brown circles, are ligands that have been observed to bind to the corresponding binding site. **c** The protein binding site (here: Janus Kinase 2, PDB-ID 6VNK[113]) is represented as a three-dimensional (3D) graph, i.e., $\mathcal{G} = (\mathcal{V}, \mathcal{E}, \mathcal{R})$ where $\mathcal{G}$ denotes the graph, $\mathcal{V}$ vertices, $\mathcal{E}$ edges and $\mathcal{R}$ the position in 3D space. All protein atoms farther away than 5 Å from any atom of the bound ligand were removed,

yielding a pocket-centric representation of the binding pocket. **d** The ligands are represented as two-dimensional (2D) graphs, i.e., $\mathcal{G} = (\mathcal{V}, \mathcal{E})$. **e** In the proposed approach, the node features within the graph are updated through a message passing process. This can be done using either 2D or 3D message passing, depending on the nature of the molecular representation. As a result of the subsequent pooling process, a latent space vector is obtained, which captures the essential characteristics and representations of the molecule. This condensed representation provides a compact encoding of the molecule's features, enabling downstream analysis, prediction, or structure generation tasks. The latent space vector can be optionally concatenated with a wishlist of desired physicochemical properties for the output molecule. This allows for the incorporation of project-specific property constraints or objectives in the de novo molecular design process. MLP denotes Multilayer Perceptron, RNN denotes Recurrent Neural Network, and LSTM refers to a type of RNN with Long Short-Term Memory cell architecture. **f** Workflow of the presented study including DRAGONFLY validation, DRAGONFLY application to peroxisome proliferator-activated receptor (PPAR), chemical synthesis and biological characterization. ADME denotes Absorption, Distribution, Metabolism, and Excretion, and FEP denotes Free Energy Perturbation calculations.

(XGBoost) (SI2.2). These results indicate that the developed models achieved a high level of accuracy in predicting the inhibition constant of novel molecules within similar domains of applicability for the targets studied. Furthermore, the performance of the ECFP and CATS models exhibited a logarithmically decreasing error as the training set size increased. Beyond a certain data set size (-100 molecules), the performance of the USRCAT models reached a plateau, indicating that additional data did not improve their predictive accuracy (Fig. 2b, Fig. S3). These findings underscore the effectiveness of utilizing a combination of descriptors, incorporating both, structural and fuzzy features, in the performance of the KRR models.

## DRAGONFLY outperforms standard chemical language models for molecular design
The evaluation criteria, which encompassed synthesizability, novelty, and predicted bioactivity were applied to evaluate virtual libraries generated de novo (Methods for details on metrices). This allowed for a comparison between DRAGONFLY and fine-tuned recurrent neural networks (RNNs). To conduct the comparison, five known ligands each were selected as templates for twenty well-studied macromolecular targets, including nuclear hormone receptors and kinases with over 200 known ligands (Tables S2–S3). DRAGONFLY demonstrated superior performance over the fine-tuned RNNs across the majority of

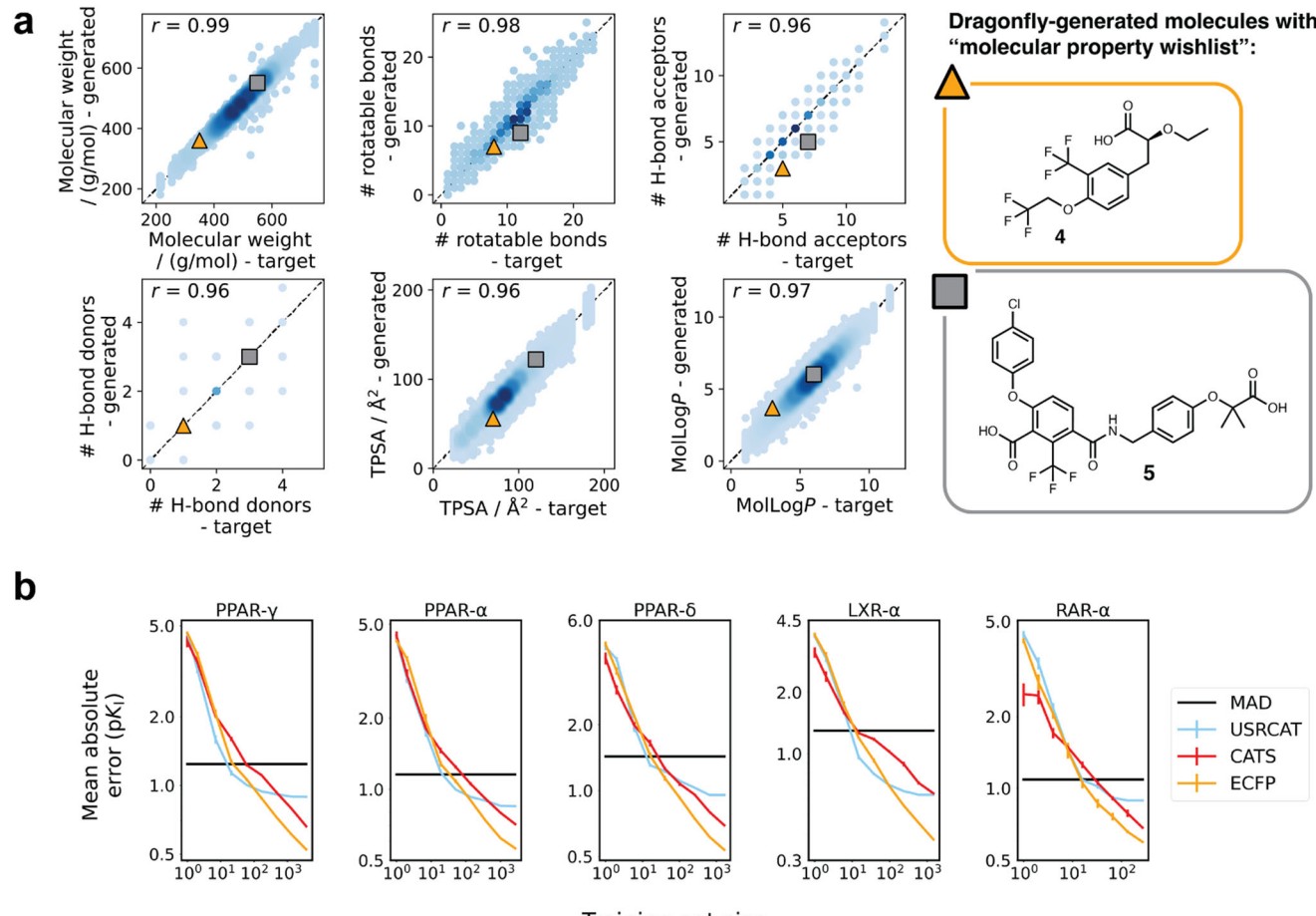

**Fig. 2 | Property translation with DRAGONFLY and quantitative structure-activity relationship (QSAR) models. a** The scatter plots depict the translation of desired properties into the generated molecules, with squared Pearson correlation coefficients ($r^2$) >0.95 for all investigated molecular properties. The high correlation suggests a strong relationship between the desired properties and the generated molecules. Properties include molecular weight, rotatable bonds, hydrogen bond acceptors, hydrogen bond donors, polar surface area, and lipophilicity calculated as MolLog*P* values[33]. Two exemplary molecules **4** and **5** and their position in the six scatter plots are visualized, showing that the desired molecular properties are accurately represented in the generated molecules. **b** The double-logarithmic learning curves depict how the prediction error of ligand-based QSAR models for three descriptors (extended connectivity fingerprint (ECFP4)[36], chemical advanced template search (CATS, absolute values)[37], ultra-fast shape Recognition with atom types (USRCAT)[38]) varies with the size of the data set, focusing on five selected drug targets related to nuclear hormone receptors. Each plot displays a horizontal line representing the mean absolute deviation (MAD) of the training data. The error bars on the plots represent the standard deviation observed during a 10-fold cross-validation. Source data are provided as a Source Data file.

templates and properties examined (Table 1, Tables S4–S6). Furthermore, using the same evaluation criteria, ligand-based design was compared to structure-based design, with ligand-based design applications outperforming structure-based models in all investigated scenarios (Table 2, Tables S4–S6).

To evaluate the potential of DRAGONFLY to generate molecules that extend into new areas of chemical space, we analyzed the similarity of the generated molecules to both the training data set (i.e., a subset of ChEMBL[28]) and an external data set (i.e., PubChem[39], excluding ChEMBL molecules). Our analysis demonstrated that the generated molecules were as similar to the molecules in PubChem when compared to the molecules in ChEMBL (Table S7). While there is a degree of similarity to known molecules, DRAGONFLY also produced a large proportion of molecules with high novelty scores and diverse structures generating higher structural and scaffold novelty than the well-established fine-tuned RNN methods (Table 1). These results suggest that DRAGONFLY is not limited to recapitulating the training data but also has the capacity to explore and generate molecules in previously uncharted regions of chemical space, albeit the extent of this exploration warrants further investigation.

We compared the performance of DRAGONFLY models trained on two widely used chemical alphabets, SMILES-strings[3] and self-referencing embedded strings (SELFIES)[40], to quantify the differences. By employing both string representations in structure- and ligand-based de novo design, we were able to directly compare their performance across various molecular properties (Table 2, Fig. S9). The DRAGONFLY models trained on SELFIES yielded a higher fraction of novel molecules among all of the 20 investigated applications (99.7 ± 0.1% vs. 92.2 ± 0.4%, Table 2) with a greater scaffold diversity (86 ± 1% vs. 53 ± 2%, Table 2) while retaining comparable structural diversity (98.8 ± 0.1% vs. 97.9 ± 0.1%, Table 2). However, the DRAGONFLY models trained on SMILES-strings more accurately fulfilled the property requirements, such as greater synthesizability (93.4 ± 0.6% vs. 84 ± 1%, Table 1), predicted bioactivity (e.g., MAE = 34.7 (± 0.3) vs 31.9 (± 0.1) for PPARγ, Table 1), as well as slightly lower mean absolute errors for physical and chemical properties (e.g., MAE = 0.027 ± 0.005 vs 0.230 ± 0.007 for hydrogen bond donors, Table 3). Overall, the use of the two chemical alphabets resulted in comparable numbers of molecules that were predicted to fulfill all desired properties. Because of the better performance of the SMILES-based models for the objectives of

**Table 1 | Comparison of DRAGONFLY with a fine-tuned recurrent neural network (RNN) approach, assessing the percentage of molecules meeting various criteria: (i) Unique and novel, (ii) Novelty score ≥ 0.65, (iii) Retrosynthetic accessibility score (RAScore) ≥ 0.5, (iv) QSAR score ≤ 1 μM, and (v) meeting all four criteria**

| Template / Method | Unique and novel / % | Novelty score ≥ 0.65 / % | RAScore ≥ 0.5 / % | QSAR score ≤ 1 μM / % | All criteria / % |
|---|---|---|---|---|---|
| **PPARγ** | | | | | |
| RNN-SMILES | 75.4 (± 2.7) | 28.7 (± 1.1) | 67.9 (± 2.3) | 29.6 (± 2.4) | 5.1 (± 0.2) |
| DRAGONFLY-SMILES | 91.8 (± 0.3) | 47.9 (± 1.4) | **86.0 (± 0.3)** | **34.7 (± 0.3)** | 9.4 (± 0.0) |
| DRAGONFLY-SELFIES | **99.8 (± 0.1)** | **77.4 (± 0.1)** | 82.2 (± 0.2) | 31.9 (± 0.1) | **13.3 (± 0.0)** |
| **LXRβ** | | | | | |
| RNN-SMILES | 92.4 (± 2.5) | 65.9 (± 2.6) | 87.9 (± 2.8) | 28.6 (± 0.9) | 11.3 (± 0.4) |
| DRAGONFLY-SMILES | 94.3 (± 0.5) | 80.2 (± 1.2) | **89.1 (± 0.5)** | 26.2 (± 0.2) | **11.8 (± 0.1)** |
| DRAGONFLY-SELFIES | **100 (± 0.0)** | **91.3 (± 0.5)** | 84.2 (± 0.3) | **27.9 (± 0.2)** | 11.1 (± 0.1) |
| **RARα** | | | | | |
| RNN-SMILES | 69.7 (± 5.9) | 41.9 (± 3.3) | 57.2 (± 4.3) | 30.1 (± 1.8) | 11.1 (± 0.7) |
| DRAGONFLY-SMILES | 92.2 (± 0.4) | 62.4 (± 0.7) | 75.6 (± 0.5) | **32.4 (± 0.7)** | 12.7 (± 0.2) |
| DRAGONFLY-SELFIES | **99.8 (± 0.0)** | **87.5 (± 0.3)** | **77.1 (± 0.2)** | 29.6 (± 0.3) | **14.0 (± 0.1)** |
| **BRAF** | | | | | |
| RNN-SMILES | 89.2 (± 3.5) | 35.1 (± 3.1) | 85.9 (± 3.0) | 35.0 (± 1.3) | 6.7 (± 0.3) |
| DRAGONFLY-SMILES | 87.9 (± 0.6) | 46.0 (± 0.8) | **80.9 (± 0.5)** | **42.9 (± 0.5)** | 10.7 (± 0.1) |
| DRAGONFLY-SELFIES | **99.7 (± 0.1)** | **81.1 (± 0.6)** | 77.3 (± 0.4) | 34.3 (± 0.1) | **12.4 (± 0.0)** |
| **BTK** | | | | | |
| RNN-SMILES | 82.0 (± 4.4) | 64.5 (± 4.1) | 61.9 (± 4.7) | 20.7 (± 1.8) | 4.5 (± 0.2) |
| DRAGONFLY-SMILES | 88.9 (± 0.7) | 53.2 (± 0.4) | 69.6 (± 0.9) | **36.3 (± 0.7)** | **8.8 (± 0.1)** |
| DRAGONFLY-SELFIES | **100 (± 0.0)** | **85.8 (± 0.7)** | 68.2 (± 1.0) | 25.8 (± 0.1) | 5.8 (± 0.0) |
| **JAK2** | | | | | |
| RNN-SMILES | 88.8 (± 3.9) | 60.2 (± 4.2) | 79.9 (± 3.4) | 35.0 (± 2.2) | 14.5 (± 0.8) |
| DRAGONFLY-SMILES | 84.8 (± 1.0) | 39.4 (± 0.9) | 69.0 (± 1.0) | **55.9 (± 1.5)** | 14.8 (± 0.2) |
| DRAGONFLY-SELFIES | **99.2 (± 0.0)** | **73.3 (± 0.8)** | **70.5 (± 0.5)** | 50.5 (± 1.0) | **18.3 (± 0.2)** |

Bold indicates whether the SELFIES- or SMILES-based models achieve a higher value for the investigated property in both structure- and ligand-based models. The values are presented as mean and standard deviation, based on three runs (*N* = 3), each sampling 2000 SMILES-strings. The complete list of 20 investigated targets can be found in Tables S2–S6. *JAK* Janus kinase, *PPAR* Peroxisome proliferator-activated receptor, *BRAF* Serine/threonine-protein kinase B-Raf (rapidly accelerated fibrosarcoma), *BTK* Bruton's tyrosine kinase, *RAR* Retinoic acid receptor, *LXR* Liver X receptor.

**Table 2 | Comparison of four Dragonfly methods, namely ligand-SMILES, ligand-SELFIES, structure-SMILES, and Structure-SELFIES**

| DRAGONFLY method | Valid and unique molecules / % | Valid, unique and novel molecules / % | RAScore ≥ 0.5 / % | Average Jaccard distance to other molecules |
|---|---|---|---|---|
| Ligand-SMILES | 93.3 (± 0.4) | 92.2 (± 0.4) | **93.4 (± 0.6)** | 0.778 (± 0.001) |
| Ligand-SELFIES | **99.9 (± 0.1)** | **99.7 (± 0.1)** | 84.0 (± 1.0) | **0.805 (± 0.002)** |
| Structure-SMILES | 90.2 (± 0.8) | 87.4 (± 0.9) | **90.0 (± 1.0)** | 0.773 (± 0.004) |
| Structure-SELFIES | **99.9 (± 0.1)** | **99.6 (± 0.1)** | 78.0 (± 2.0) | **0.811 (± 0.003)** |
| | **Unique atom scaffolds / %** | **Unique and Novel atom scaffolds / %** | **Unique carbon scaffolds / %** | **Unique and novel carbon scaffolds / %** |
| Ligand-SMILES | 85.0 (± 0.1) | 53.0 (± 0.2) | 98.4 (± 0.3) | 58.0 (± 0.2) |
| Ligand-SELFIES | **96.9 (± 0.4)** | **86.0 (± 0.1)** | **99.8 (± 0.1)** | **83.0 (± 0.1)** |
| Structure-SMILES | 84.0 (± 0.1) | 55.0 (± 0.3) | 98.3 (± 0.3) | 56.0 (± 0.2) |
| Structure-SELFIES | **96.0 (± 0.1)** | **81.0 (± 0.1)** | **99.9 (± 0.1)** | **83.0 (± 0.2)** |

Bold indicates whether SELFIES- or SMILES-based models achieve a higher value for the investigated property in both structure- and ligand-based models. The percentage of molecules is shown that fulfill the desired criteria: (i) valid and unique molecules, (ii) valid, unique, and novel molecules, (iii) fraction of molecules with an RAScore of ≥ 0.5, (iv) average Jaccard distance to other generated molecules from the same run (indicating diversity), and (v)–(viii) various scaffold metrics, including unique and novel carbon and atom scaffolds. The values are presented as mean and standard deviation, based on three Dragonfly runs (*N* = 3), each sampling 2000 SMILES-strings.

synthetic accessibility, bioactivity, and desired physical and chemical properties, we used these models in the prospective study.

**Structure-based design with DRAGONFLY generates potential novel ligands**

DRAGONFLY was utilized in a prospective manner for structure-based ligand design targeting human PPARγ (PPARγ, PDB-ID 3G9E[41]). The nuclear hormone receptor PPARγ is one of the three peroxisome proliferator-activated receptors (i.e., PPARγ/α/δ), that have been exploited as drug targets for combating multiple diseases, in particular metabolic syndrome-related disorders and cancer[42–44]. Activation of the PPARs by natural ligands or by synthetic PPAR agonizts triggers the formation of heterodimers with members of the retinoid X receptor (RXR) family[45]. Upon recruitment of specific cofactors, these

**Table 3 | Accuracy of the desired physical and chemical properties of molecules generated by DRAGONFLY**

| DRAGONFLY method | MW | Rot. B. | HBA | HBD | PSA | LogP |
|---|---|---|---|---|---|---|
| Unit | g mol$^{-1}$ | # | # | # | Å$^2$ | – |
| **MAD** | 75.52 | 2.81 | 0.981 | 1.69 | 27.08 | 1.25 |
| **Ligand-SMILES** | | | | | | |
| MAE | 7.7 (± 0.2) | 0.29 (± 0.01) | 0.23 (± 0.01) | 0.027 (± 0.005) | 4.4 (± 0.2) | 0.252 (± 0.004) |
| MAD / MAE | **9.8 (± 0.2)** | **9.9 (± 0.48)** | **4.3 (± 0.21)** | **63 (± 11)** | **6.1 (± 0.2)** | **4.94 (± 0.08)** |
| **Ligand-SELFIES** | | | | | | |
| MAE | 8.0 (± 0.2) | 0.88 (± 0.040) | 0.40 (± 0.014) | 0.230 (± 0.007) | 6.8 (± 0.21) | 0.380 (± 0.006) |
| MAD / MAE | 9.4 (± 0.2) | 3.2 (± 0.14) | 2.5 (± 0.09) | 7.3 (± 0.3) | 4.0 (± 0.1) | 3.27 (± 0.05) |
| **Structure-SMILES** | | | | | | |
| MAE | 12.1 (± 0.5) | 0.42 (± 0.02) | 0.28 (± 0.02) | 0.046 (± 0.007) | 4.6 (± 0.1) | 0.315 (± 0.008) |
| MAD / MAE | **6.2 (± 0.3)** | **6.7 (± 0.3)** | **3.5 (± 0.2)** | **37 (± 6)** | **5.9 (± 0.2)** | **4.0 (± 0.1)** |
| **Structure-SELFIES** | | | | | | |
| MAE | 15 (± 0.4) | 1.12 (± 0.04) | 0.50 (± 0.03) | 0.27 (± 0.02) | 7.4 (± 0.3) | 0.426 (± 0.008) |
| MAD / MAE | 5.03 (± 0.1) | 2.5 (± 0.09) | 2.0 (± 0.1) | 6.3 (± 0.4) | 3.6 (± 0.1) | 2.92 (± 0.05) |

Bold indicates if the SELFIES- or the SMILES-based models achieved a higher value for the investigated property. Abbreviations: *MAD* Mean absolute deviation, *MAE* Mean absolute error, *MW* Molecular weight, *Rot. B.* Number of rotatable bonds, *HBA* Hydrogen bond acceptors, *HBD* Hydrogen bond donors, *PSA* Polar surface area. The numbers are presented as the mean and standard deviation, with a sample size of $N = 3$, i.e., 3 DRAGONFLY runs, each sampling 2000 SMILES-strings. MAD / MAE yields a number that indicates by which factor a model is better than the MAD.

heterodimers transactivate PPAR-responsive elements (PPREs) of target genes involved in insulin signaling, lipid and glucose metabolism, immune response, as well as cell cycle and differentiation[46,47]. Several activators with different selectivity for the respective PPAR subtypes have reached advanced clinical trials or were introduced to the market. Aiming to test the method's ability to generalize, the training of DRAGONFLY did not include the protein template PPAR or any closely related structures present in the training data set. Specifically, proteins belonging to the same sub-family (PPARα, PPARγ, PPARδ) or other species were intentionally excluded. The most closely related proteins in the training data were found to be thyroid hormone receptor β-1 and liver X receptor (LXR) β, sharing a sequence identity of 33% and 30%, respectively, with PPARγ (SI8, Table S8).

To obtain a library of candidate ligands, the ligand-binding site of human PPARγ protein (PDB-ID 3G9E[41]) was utilized as the structural template (Fig. 3a). A total of 300 k molecules were generated, and subsequent filtering was performed based on specific criteria. These filters included an upper molecular weight limit of ≤600 g mol$^{-1}$, a RAScore threshold of ≥0.5, and a novelty score of ≥0.7. The resulting subset of molecules obtained from the filtering process was further ranked using KRR-based QSAR scoring based on the average predicted binding affinity ($pK_I$ or $pIC_{50}$), combining the ECFP (double weighted contribution), CATS, and USRCAT descriptors. Aiming to explore the potential of the computer-generated molecules for dual-target activity and receptor selectivity, two different scoring procedures were prospectively evaluated. The first procedure focused on selective single-target affinity to PPARγ. The second procedure involved assigning equal weights to dual-target affinity towards both PPARγ and PPARδ. The decision to focus on affinity towards PPAR sub-family members was made to align with their clinical significance[48]. Applying these ranking criteria led to the identification of twice the top-5 molecules (**1, 2, 6-13**), depicted in Fig. 3b.

While exhibiting sufficient structural and scaffold novelty, the generated molecules also possess a topology commonly observed in modulators of the PPAR subfamily, as captured by the three different QSAR models. Specifically, they feature an acidic head group connected to an aromatic core through a linker. Furthermore, this core is linked to another single- or bicyclic aromatic ring system. While the propionic acid head group was predominant among the top-ranking designs, it is worth noting that various other head groups were also present among the 100 highest-scored de novo molecules. Figure 3c highlights a selection of non-carboxylic head groups and secondary amides from this top-100 set. This selection includes a diverse range of secondary amides as well as pyrimidine-diones, i.e., head groups known to promote PPARγ modulation[49,50]. These alternative head groups demonstrate the structural diversity and potential for exploring different chemistries and bioisosters in the design of novel molecules within the top-ranked subset.

## Molecules generated with DRAGONFLY potently and selectively activate PPARγ

To test the practical applicability and usefulness of the structure-based molecular design algorithm, the two top-scoring de novo generated designs (**1** and **2**) were chosen for chemical synthesis and subsequent biological characterization. Design **1** was achieved through a convergent synthesis comprising a total of 10 steps, with the longest sequential route consisting of six steps, with an overall yield of 12% (Fig. 4a). Design **2** was synthesized through five steps achieving an overall yield of 0.6% (Fig. 4b). Additionally, regioisomer **3** was isolated during the synthesis of design **2**.

Subsequent biological testing of the three molecules (**1**–**3**) in a cell-based reporter gene assay confirmed the intended activity profiles (SI9). Compound **1** exhibited the desired and predicted dual activity on PPARγ and PPARδ at half maximal effective concentration $EC_{50}(PPARγ) = 1.5 ± 0.2 \mu M$, and $EC_{50}(PPARδ) = 0.24 ± 0.05 \mu M$, respectively (Fig. 5a, Fig. S13). Moreover, compound **1** was characterized for its affinity to the ligand binding domain of PPARγ by isothermal titration calorimetry (ITC), yielding a measured dissociation constant of $K_D = 0.8 ± 0.1 \mu M$, and a molar ratio of one ligand per protein molecule (Fig. 5b). This $K_D$ value of compound **1** confirmed the observed direct receptor modulation in the functional reporter gene assay. Compound **2** demonstrated a noteworthy level of selective activity on PPARγ with an $EC_{50}$ value of $2.3 ± 0.7 \mu M$, while displaying no discernible impact on PPARα or PPARδ (Fig. 5c, Fig. S13). This outcome aligns seamlessly with the intended design objective. Compound **3** exhibited a dual, partial agoniztic activity profile, acting on both PPARγ (with an $EC_{50}$ of $1.8 ± 0.1 \mu M$) and PPARα (with an $EC_{50}$ of $3.4 ± 0.3 \mu M$), while showing no discernible activity towards PPARδ. Furthermore, the predicted selectivity of compounds **1**–**3** towards other nuclear hormone receptor targets was experimentally confirmed for retinoid X receptor (RXR)α, liver X receptor (LXR)α, and retinoic acid receptor (RAR)α (Fig. 5d).

Computer-designed compounds **1** and **2** underwent initial in vitro testing to assess their absorption, distribution, metabolism, and

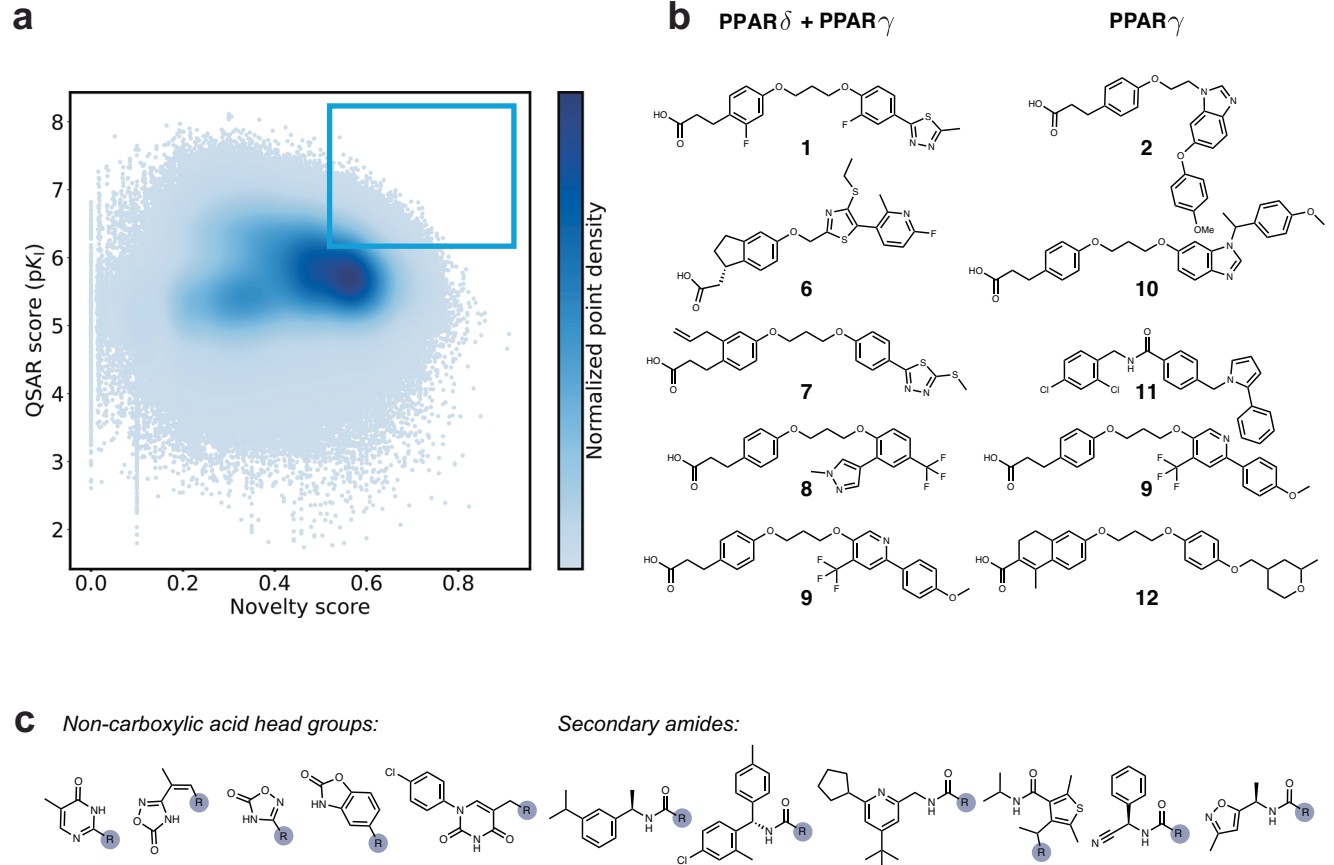

**Fig. 3 | Results of structure-based de novo design with DRAGONFLY. a** The scatter plot presented showcases the molecules designed de novo by utilizing the human peroxisome-proliferator-activated-receptor (PPAR)γ binding pocket as a template (PDB-ID 3G9E[41]). The plot displays the quantitative structure-activity relationship (QSAR) score representing the predicted binding affinity to PPARγ against the novelty score. The desired region for the generated molecules, which satisfies both, novelty and the predicted bioactivity requirements, is highlighted by a blue box located in the upper right corner of the plot. **b** Molecular structures of the top-ranking de novo designs. The predicted quality of the molecules decreases progressively from top to bottom. The ranking criteria were PPARγ + PPARδ dual-target affinity and structural novelty (left, **1** & **6**–**9**), or PPARγ single-target affinity and structural novelty (right, **2** & **9**–**12**). Molecule **9** ranks among the top five molecules on both lists. **c** Examples of non-carboxylic head groups and secondary amides from the top-100 molecules ranked for PPARγ, where the gray shaded *R* group represents an aromatic moiety connected to a linker. Source data are provided as a Source Data file.

excretion (ADME) properties, and were compared to the phase three dual PPARα/γ co-agonist aleglitazar[41] (SI11, Table S14). Both ligands exhibited lipophilicity values within the range of 1–2 (log*D***1**: 1.5 **2**: 1.7), which falls within the preferred range for orally administered drugs and is comparable to the log*D* value of aleglitazar (log*D*: 1.4).

In terms of permeability through membranes, both molecules displayed favorable results in the parallel artificial membrane permeability assay (PAMPA), with permeation coefficients (PAMPA$_{PEFF}$) of 3.9 cm · $s^{-1}$ · $10^{-6}$ for compound **1** and 14 cm · $s^{-1}$ · $10^{-6}$ for compound **2**. Achieving sufficient cell permeability is crucial for targeting the PPARγ receptor, located within the cell nucleus. Cellular permeability was confirmed in the P-glycoprotein (Pgp) efflux assay for compounds **1** and **2**, revealing values of 1.6 (15 nm · $sec^{-1}$) and 1.2 (60 nm · $sec^{-1}$), respectively. The observed efflux ratio indicates that compounds **1** and **2** are only interacting weakly with the Pgp transporter and thus, hold high potential to reach multiple cell and tissue types following a systemic application. Moreover, the unbound fractions of compounds **1** and **2** were determined at 0.42% and 0.21%. Such low unbound fractions are attributed to the negatively charged carboxylic acid, similar to other drug-like molecules containing carboxylic acid groups[41]. Furthermore, the clearance values of compounds **1** and **2** in human, rat, and mouse microsomes were consistently low (≤10 µL · $min^{-1}$ · $mg^{-1}$ protein) when compared to aleglitazar, suggesting the potential for

achieving high oral bioavailability in both humans and rodents for efficacy studies. Compound clearance rates in human hepatocytes were determined at 19 µL · $min^{-1}10^6cells^{-1}$ for both compounds **1** and **2** (Table S14). Both metabolic and hepatocyte clearance suggest a sufficient metabolic profile, paving the way for further in vivo pharmacokinetic studies. Compounds **1** and **2** exhibited no interaction with the seven pivotal cytochrome P450 isoenzymes (CYP)−Cyp3A4, Cyp1A2, Cyp2B6, Cyp2C9, Cyp2D6, Cyp2C19, and Cyp2C8 - at dose-response experiments up to 20 µM (Table S15). Moreover, both compounds presented a favorable profile in an expansive panel screen assessing multiple safety-critical off-targets. Importantly, none of the targets exhibited binding or inhibition above 60% at a compound concentration of 10 µM (Tables S13–S16). Furthermore, compounds **1** and **2** did not indicate any cytotoxicity on HEK293T cells at different time points as well as a broad range of cell numbers and compound concentrations (Fig. 5e, Fig. S16). Collectively, the computer-designed compounds **1** and **2** showcase a promising drug-like profile, signifying substantial potential for advancement in further drug development.

To investigate the binding pose of compound **1**, X-ray structure determination of the ligand-protein complex with PPARγ was conducted (Fig. 6a, SI10). Compound **1** was bound to one of two protein molecules in the asymmetric unit. Moreover, the observed binding pose showed how the relevant structural motifs of the design contribute to

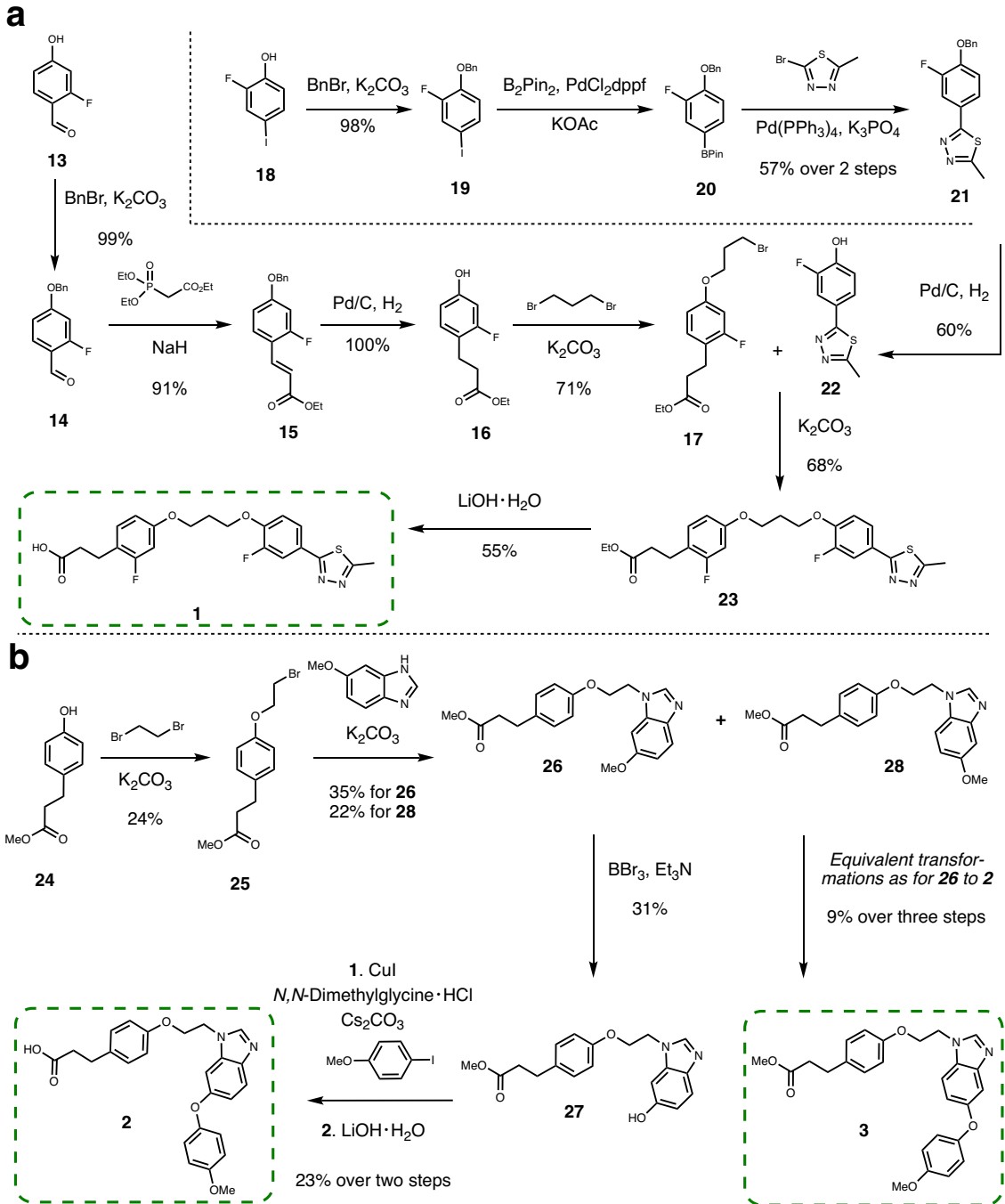

**Fig. 4 | Chemical synthesis of compounds 1, 2 and 3. a** The synthesis of compound **1** employed a convergent approach, starting from commercially available building blocks **13** and **18**, and spanning a total of 10 steps. The longest sequential route involved 6 steps. The overall yield achieved for the synthesis of **1** was 12%. **b** For the synthesis of compounds **2** as well as its regioisomer **3**, the starting material used was a commercial building block **24**. These compounds were synthesized through a sequential five-step synthesis. The overall yield obtained for compound **2** was 0.6%, while compound **3** was isolated with a yield of 0.5%. For details see SI13.

ligand-receptor interaction. Compound **1** bound in the orthosteric site lined by helices H3 and H11. The buried propionic acid head group is engaged in four intermolecular hydrogen bridges. Three of them are established with the side chains of Tyr[473], His[323], and Ser[289], whereas the fourth one is a water-mediated hydrogen bond with residue His[449]. In the empty PPARγ site, the carboxyl C-terminus of TYR[477] is blocking the site by binding in a similar position as the propionic acid head of the ligand. The ligand's tail moiety is exposed to the solvent, and the propylene glycol-like linker allows the ligand to enter the hydrophobic part of the binding pocket, where the two aromatic ring systems engage in additional interactions with the protein (Fig. 6a).

To computationally assess the binding of compounds **1** and **2** to PPARγ, absolute protein-ligand binding free-energy perturbation (ABFEP) calculations were carried out[51,52]. Compounds **1** and **2** as well as different ligands from ChEMBL with known sub-micromolar PPARγ activity (ChEMBL IDs: ChEMBL391987, ChEMBL241299, ChEMBL213355, ChEMBL212591) were modeled into the PPARγ-aleglitazar X-ray crystal structure (PDB ID: 3G9E)[41]. Compounds **1** and **2** have calculated Δ Gibbs Free Energy ($\Delta G$) values of -20.1 kcal·mol⁻¹ and -19.7 kcal·mol⁻¹, respectively. These values are in the range of other known PPARγ ligands with sub-micromolar activity, further supporting the relevance of the proposed molecules for PPAR modulation (Table 4, Fig. S17).

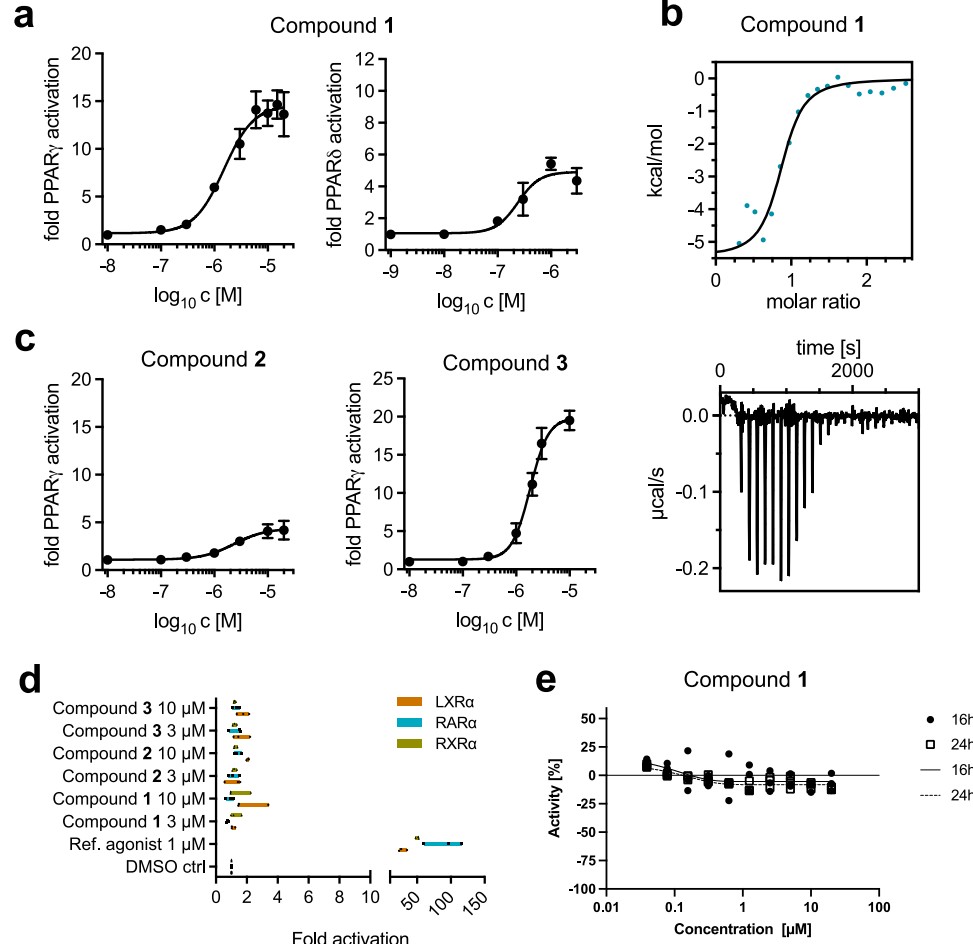

**Fig. 5 | Biological characterization of compounds 1–3.** The graphs in panels **a** and **c** display a nonlinear fitting curve derived from the mean values of three measurements ($N = 3$). The graph in panel **d** shows mean values of three measurements ($N = 3$). Error bars, represented by whiskers, indicate the standard deviations. **a** Peroxisome proliferator-activated receptor (PPAR) activation by compounds **1, 2** and **3** using a hybrid reporter gene assay. **b** Result of an isothermal titration calorimetry experiment ($N = 2$) measuring direct binding of compound **1** to PPAR$\gamma$. **c** Dose-response curves from a hybrid reporter gene assay for compounds **2** and **3** measuring PPAR$\gamma$ activation. **d** Receptor selectivity of compounds **1, 2**, and **3** for activation of liver X receptor (LXR)$\alpha$, retinoic acid receptor (RAR)$\alpha$, and retinoid X receptor (RXR)$\alpha$. **e** Cytotoxicity of compound **1** on HEK293T cells for two time points (i.e., 16 h and 24 h), 10 different concentrations (i.e., 0.05–20 $\mu$M) and 10000 cells · well$^{-1}$ ($N = 3$). Scale reference: The axes are scaled through neutral control (i.e., Dimethylsulfoxid [DMSO], set to 0) and inhibitor control wells (i.e., 20 $\mu$M Staurosporine[114], set to −100). Source data are provided as a Source Data file.

## Discussion

The generative deep learning method referred to as DRAGONFLY was evaluated in the context of ligand-based and structure-based molecular design tasks. The collective results specifically highlight the success of structure-based de novo design of potent partial agonizts for PPAR$\gamma$. These molecules effectively interact with the receptor in a canonical binding mode, while also demonstrating the desired selectivity towards the receptor and favorable ADME properties.

By leveraging an interactome-based deep learning approach and employing a graph-to-sequence neural network architecture, DRAGONFLY addresses certain challenges commonly encountered in generative molecular design methods. This approach demonstrated to (i) achieve similar or even superior results compared to a respectively fine-tuned RNN-based CLM for drug-like ligand templates, (ii) enable structure-based design using 3D protein binding sites, and (iii) effectively incorporate desired physical and chemical properties into the generated molecules. Its ability to combine structure-based and ligand-based approaches, as well as its capacity to incorporate desired properties makes it a potentially useful tool for medicinal chemistry.

The design algorithm has demonstrated its capability to successfully generate molecules with desired properties by incorporating an additional encoding within the input. This encoding allows for the translation of various drug discovery-relevant properties with high accuracy into the generated molecules. Properties such as molecular weight, the number of rotatable bonds, hydrogen-bond acceptors, hydrogen-bond donors, polar surface area, and lipophilicity can be effectively encoded and incorporated into the molecular design process. This means that the algorithm can generate molecules that not only possess the desired structural characteristics but also meet specific physical and chemical property requirements. The ability to accurately translate these user-defined properties into the generated molecules is a potentially substantial advantage of the approach. It enables researchers to identify novel molecules with specific properties and optimize them for desired therapeutic effects, bioavailability, and safety profiles. In an initial assessment, the top-ranking computer-generated molecules revealed favorable in vitro ADME properties.

The results of the study also indicate that ligand-based de novo design outperformed structure-based models for the majority of investigated molecular properties. This performance difference could be attributed to the complexity of the input and the availability of training data. Whereas a small-molecule graph typically comprises up to 200 edges describing covalent bonds, protein binding sites represented by 3D graphs are considerably larger with an average scaling factor of 60. Furthermore, the ligand-based data set used in the study

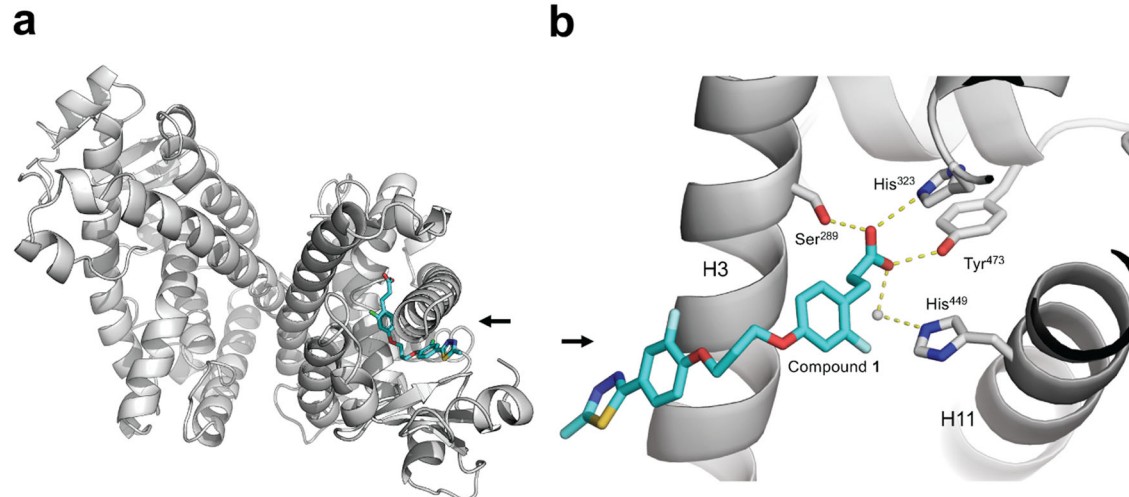

**Fig. 6 | Protein-ligand co-crystallization.** Crystal structure complex of the ligand-binding domain of human nuclear receptor PPARγ and compound **1**. (**a**) Illustration of the asymmetric unit containing two protein molecules, one of which shows compound **1** bound. Refined cartoon structure of the complex showing compound **1** (stick model in blue) bound to chain A and chain B empty (chain A on the right and chain B on the left). (**b**) Close-up view of the binding pose of compound **1** (blue) with hydrogen bonds to residues Tyr[473], His[323], and Ser[289] as well as a water-mediated hydrogen bond to His[449] shown as dashed lines (PDB-ID: 8PBO). The black arrowhead marks the entry to the binding pocket of the receptor. To facilitate perceptibility, certain protein residues have been omitted from the illustration. Source data are provided as a Source Data file.

consisted of around 501 k distinct bioaffinity values, whereas only about 236 k bioaffinities were accessible to the structure-based training procedure. This disparity in training data availability may contribute to the superior performance of the ligand-based design models. Nevertheless, it is important to acknowledge that one benefit of structure-based design applications is their flexibility in not mandating exceptionally high-quality query ligands for the generation of molecules. This applicability can be valuable for in silico library design in scenarios where relevant ligand information is limited or unavailable, e.g., for newly identified disease-relevant macromolecular targets.

Such a scenario was emulated in the prospective application to PPARγ. However, it is worth noting that in this study, QSAR models were employed for scoring, and these models were trained on existing ligand activity data. The successful machine learning from the relevant training data is evident in the discovery that compound **1** interacts with the receptor in the canonical binding mode[53], as evidenced in the crystallographic complex. Further studies will be essential to combine DRAGONFLY with scoring functions not involving known active query ligands for bioaffinity assessment. Structure-based scoring for binding pose estimation, bioactivity prediction and virtual screening has indeed shown to be one of the most challenging topics in computational drug design[54]. Prominent among the existing structure-based

scoring models are free energy perturbation (FEP) techniques[51,55], geometric deep learning approaches[56–59], machine-learned force fields[60], and purely statistics-driven models[61–63], which currently receive considerable attention. Moreover, emphasis will be directed towards the utilization of DRAGONFLY applications to create bioactive ligands for protein models derived from apo protein structures (where no ligand is bound), and structure prediction methods like AlphaFold[64]. Understanding the algorithm's performance in these scenarios will provide valuable insights into its applicability and potential limitations in de novo drug design with structure-based ligand scoring and predicted protein structures.

The comparison between DRAGONFLY models trained on SMILES-strings and SELFIES demonstrated similar overall results in terms of different templates and properties. However, certain trends were observed: Libraries generated using SELFIES exhibited a higher level of diversity and novelty, while libraries generated using SMILES-strings achieved higher accuracy in incorporating desired molecular properties such as synthesizability, physical and chemical properties, or predicted bioactivity. These findings unveil specific strengths and trade-offs associated with de novo design approaches based on SELFIES and SMILES strings. Both methodologies have their advantages and limitations, and the choice between them depends on specific requirements and available data.

The combination of DRAGONFLY with scoring functions incorporating compound synthesizability, novelty, and bioactivity towards one or multiple targets using various descriptors was demonstrated to be feasible. By applying a tailored combination of these properties, we achieved promising results, i.e., in the exemplary case of PPARγ. The approach allowed for the identification of top-ranking molecules that exhibited sufficient structural and scaffold novelty, synthesizability, and a desired bioactivity profile across multiple targets. While molecular novelty has been extensively discussed in recent literature, most studies mainly focus on descriptor similarity using structural fingerprints[65,66]. We showed that incorporating additional scaffold criteria can enhance the novelty of top-ranking molecules. By assigning higher weights to different descriptors, distinct outcomes were observed. When higher weights were assigned to the ECFP fingerprint descriptor, the generated molecules exhibited higher structural similarity to known ligands. This approach favored the exploration of the

## Table 4 | Absolute protein-ligand binding free-energy perturbation calculations

| ChEMBL ID | ΔG / kcal · mol⁻¹ | pEC50 |
|---|---|---|
| Compound **1** | -20.1 | 5.83 |
| Compound **2** | -19.7 | 5.64 |
| ChEMBL391987 | -22.3 | 6.94 |
| ChEMBL241299 | -18.4 | 6.51 |
| ChEMBL213355 | -19.1 | 6.82 |
| ChEMBL212591 | -19.7 | 6.64 |

The numbers represent the calculated absolute protein-ligand binding free-energy perturbation (ABFEP, ΔG) for compounds **1** and **2**, and ligands from the ChEMBL database with known sub-micromolar PPARγ activity (database entries ChEMBL391987, ChEMBL241299, ChEMBL213355, ChEMBL212591). The ABFEP calculations were executed with default parameters over a simulation period of 5 ns. (see Methods).

known chemical space and enabled the design of molecules with recognizable structural features. In contrast, assigning higher weights to the two "fuzzy" descriptors, CATS and USRCAT, resulted in top-ranking molecules that deviated further from the structurally explored space, as expected[67,68]. This latter approach focused on prioritizing novelty and divergence from known ligands, allowing for the exploration of new chemical territories. These findings highlight the flexibility and versatility of the scoring functions and the ability to customize the weights of different descriptors to achieve specific objectives. By balancing the importance of structural similarity, novelty, and other desired properties, we were able to guide the generation of molecules that met the desired criteria for the targeted application, such as PPARγ in this case.

The chemical synthesis of two top-ranking de novo designs, designated as compounds **1** and **2**, along with regioisomer **3**, turned out to be comparably cumbersome, requiring 10 and 5 synthesis steps, respectively. This observation together with low yields point to limitations of the employed scoring function for synthesizability, motivating the development of better measures safeguarding straightforward synthesis of molecules designed with a generative models. For example, a hybrid structure generator for DRAGONFLY that combines rule-based molecule selection with predictive deep learning models could be envisaged[69,70].

Subsequent biological evaluation of compounds **1**–**3** was carried out using the human PPARγ ligand-binding domain. Both, cell-based activity and direct binding, could be confirmed. These investigations led to the identification of novel PPAR modulators that exhibited low micromolar to high nanomolar activity. Importantly, the intended behavior and specificity characteristics for which the two designs were originally prioritized received confirmation through experimental verification. These results highlight their ability to target PPAR with precision, while also sidestepping pronounced influence on closely affiliated nuclear hormone receptors, like RXR, and a sizeable panel of other undesired off-targets. This outcome demonstrates the efficacy of the structure-based DRAGONFLY de novo design approach in generating molecules with the desired properties and biological activities, including selected ADME properties. The observed lack of CYP interaction up to a compound concentration of 10 μM is a crucial aspect in averting drug-drug interactions, which is of particular relevance for the treatment of metabolic syndrome, where patients frequently require concurrent administration of multiple drugs[71]. These results represent a substantial milestone as they showcase the successful application of a generative deep learning model for molecular de novo design that incorporates ligand activity and selectivity on multiple targets, as well as panel selectivity within the same protein class, exemplified in the context of nuclear hormone receptors.

The concept of interactome-based deep learning was introduced to de novo molecular design as a means to maximize the information learned about interaction networks between drug targets and their ligands. It could be demonstrated that by employing an interactome-based training procedure, some of the limitations encountered by transfer-learning-based CLMs can be mitigated. It is important to note that the concept of interactome-based deep learning is not limited to the specific neural network architecture or drug-target graph presented in this study. There is room for exploration within various frameworks and methodologies. For instance, other neural network architectures, such as sequence-to-sequence models using transformer neural networks[72], or graph-to-graph-based architectures utilizing diffusion-based models[20], could be implemented. These variations would enable additional ligand-based design approaches. The graph-to-graph models would extend the capability to incorporate 3D information for structure-based molecular design, while sequence-to-sequence models would be limited to protein sequence information. Furthermore, the drug-target interactome could be expanded to include additional targets beyond those considered in the presented

study. For example, RNA binding sites, protein surface binders such as molecular glues or certain macrocycles, or protein-protein interactions could be included. This expansion of the interactome would enable the exploration of distinct design possibilities and target-specific applications.

In the presented context, interactome-based deep learning serves as a proof-of-concept for "zero-shot" learning that can be further adapted and customized for specific applications in small molecule drug discovery, ultimately leading to more efficient hit-and-lead discovery in bioorganic and medicinal chemistry. By leveraging data-driven deep learning and interaction networks, this approach offers new avenues for foundation models enabling tailored molecular design strategies and the discovery of innovative drug candidates.

## Methods
### Neural network architecture
The DRAGONFLY method employs a graph neural network architecture[73–75]. This approach utilizes a GTNN model to encode the input molecular graph, which is represented as a 2D graph for ligands and a 3D graph for protein binding sites. The GTNN transforms the graph into a condensed one-dimensional (1D) feature vector. Subsequently, this feature vector is decoded back into the corresponding molecular string, using a CLM based on an RNN-LSTM[32,76] architecture for the molecule generation process.

**Graph transformer neural network.** *Message passing*: The atomic features were embedded and transformed using a multilayer Perceptron (MLP) to obtain atomic feature vectors $\mathbf{h}_i^0$. Message passing as suggested by Satorras et al.[77] and used in other 3D-based prediction tasks[78,79] was applied to $L = 3$ layers, iteratively applied over all atomic representations $\mathbf{h}_i^0$. Edges were introduced differently in the 2D and 3D graph representations. In the 2D graph, edges were established between atoms connected by covalent bonds. On the other hand, in the 3D graph, edges were formed between all atoms situated within a radius of 4 Å from each other. This approach ensured that the molecular structures were accurately represented in both 2D and 3D formats, effectively capturing the most relevant interactions occurring between atoms. In each iteration of the message-passing layer, the atomic representations underwent a transformation as described by Equation (1).

$$\mathbf{h}_i^{l+1} = \phi\left(\mathbf{h}_i^l, \sum_{j \in \mathcal{N}(i)} \psi\left(\mathbf{h}_i^l, \mathbf{h}_j^l\right)\right),\tag{1}$$

for 2D graph structures, and Equation (2)

$$\mathbf{h}_i^{l+1} = \phi\left(\mathbf{h}_i^l, \sum_{j \in \mathcal{N}(i)} \psi\left(\mathbf{h}_i^l, \mathbf{h}_j^l, \mathbf{r}_{i,j}\right)\right),\tag{2}$$

for 3D graph structures.

In Equations (1) and (2) $\mathbf{h}_i^l$ is the atomic representation $\mathbf{h}$ of the $i$-th atom at the $l$-th layer; $j \in \mathcal{N}(i)$ is the set of neighboring nodes of atom $i$ connected via edges; $\mathbf{r}_{i,j}$ the inter-atomic distance represented in terms of Fourier features, using a sine- and cosine-based encoding; $\psi$ is an MLP transforming node features into message features $\mathbf{m}_{ij}$: $\mathbf{m}_{ij} = \psi(\mathbf{h}_i^l, \mathbf{h}_j^l, \mathbf{r}_{i,j})$ for 3D graphs, and $\mathbf{m}_{ij} = \psi(\mathbf{h}_i^l, \mathbf{h}_j^l)$ for 2D graphs; $\sum$ denotes the permutation-invariant pooling operator (i.e., sum) transforming $\mathbf{m}_{ij}$ into $\mathbf{m}_i$: $\mathbf{m}_i = \sum_{j \in \mathcal{N}(i)} \mathbf{m}_{ij}$; and $\phi$ is an MLP transforming $\mathbf{h}_i^l$ and $\mathbf{m}_i$ into $\mathbf{h}_i^{l+1}$. The atomic features from all layers $[\mathbf{h}_i^{l=1}, \mathbf{h}_i^{l=2}, \mathbf{h}_i^{l=3}]$ were concatenated and transformed via an MLP, resulting in final atomic features $\mathbf{H}_i$. The features $\mathbf{H}_i$ were subsequently pooled into a molecular representation via a graph multiset transformer (GMT) and further transformed via two MLPs to the two 1D latent space

representations $\mathbf{l}^1_{t=0}$ and $\mathbf{l}^2_{t=0}$. A detailed description of the GMT module can be found elsewhere[30].

**Long-short-term memory neural network.** LSTM neural networks represent a specific category of recurrent neural networks renowned for their capacity to understand and produce sequences of characters. Their proficiency in comprehending sequential data and capturing intricate temporal connections renders them suitable for de novo drug design applications. In this context, the LSTM architecture was integrated to convert the acquired hidden states from the GTNN (i.e., $\mathbf{l}t=0^1$ and $\mathbf{l}t=0^2$) into a molecule represented in string form (SMILES or SELFIES). $\mathbf{l}^1_{t=0}$ and $\mathbf{l}^2_{t=0}$ are used as the initial hidden states of the LSTM architecture. At each time step $t$ the next character of the sequence $\omega_{t+1}$ is predicted given the two hidden states $\mathbf{l}^1_t$ and $\mathbf{l}^2_t$, the two memory cell states $\mathbf{c}^1_t$ and $\mathbf{c}^2_t$, and the embedding $\mathbf{k}_t$ of the previous character in the sequence $\omega_t$. This transformation is conducted using four non-linear transformations via Equation (3):

$$
\begin{aligned}
\mathbf{g}_i &= \sigma(\mathbf{W}_{ix}\mathbf{k}_t + b_{ix} + \mathbf{W}_{il}\mathbf{l}_{t-1} + b_{il}) \\
\mathbf{g}_f &= \sigma(\mathbf{W}_{fx}\mathbf{k}_t + b_{fx} + \mathbf{W}_{fl}\mathbf{l}_{t-1} + b_{fl}) \\
\mathbf{g}_o &= \sigma(\mathbf{W}_{ox}\mathbf{k}_t + b_{ox} + \mathbf{W}_{ol}\mathbf{l}_{t-1} + b_{ol}) \\
\widetilde{\mathbf{c}}_t &= \tanh(\mathbf{W}_{cx}\mathbf{k}_t + b_{cx} + \mathbf{W}_{cl}\mathbf{l}_{t-1} + b_{cl}) \\
\mathbf{c}_t &= \mathbf{g}_f \odot \mathbf{c}_{t-1} + \mathbf{g}_i \odot \widetilde{\mathbf{c}}_t \\
\mathbf{l}_t &= \mathbf{g}_o \odot \mathbf{c}_t
\end{aligned}
\tag{3}
$$

where $\mathbf{l}_t$ and $\mathbf{c}_t$ represent the hidden state and the memory cell state at time $t$, respectively. $\mathbf{g}_i$, $\mathbf{g}_f$ and $\mathbf{g}_o$ represent the input, forget, and output gates, respectively. $\sigma$ and $\odot$ indicate the sigmoid activation function and the Hadamard product[80], respectively. $\widetilde{\mathbf{c}}_t$ represents the candidate memory cell state, which is used to update the previous memory cell state $\mathbf{c}_{t-1}$. $\mathbf{W}$ and $b$ are the weights and biases used for the corresponding linear transformations. The resulting updated hidden state $\mathbf{l}_t$ is then transformed using a softmax activation function to obtain a logit vector $\hat{\mathbf{y}}_t$ (i.e., a vector with the dimension of the alphabet $\Omega$) via Equation (4):

$$
\hat{\mathbf{y}}_t = \text{softmax}(\mathbf{W}_{yl}\mathbf{l}_t + b_{yl})
\tag{4}
$$

Throughout the training phase, the cross-entropy loss was computed based on $\hat{\mathbf{y}}t$ and the ground truth $\mathbf{y}_t$. The ground truth vector $\mathbf{y}_t$ was structured with zeros in all positions except for the character's anticipated location, which was assigned a value of 1 for each prediction in the sequence. Subsequently, this calculated loss was backpropagated seamlessly through the LSTM and GTNN networks in an end-to-end manner. The training process involved the application of teacher forcing, as described in the work by Lamb et al.[81].

**Molecule sampling**
Temperature sampling was employed as a mechanism to facilitate the generation of a diverse array of output molecules using a trained DRAGONFLY model[7], achieved through Equation (5):

$$
P(\hat{\mathbf{y}}_{t+1} = \omega | \hat{\mathbf{y}}_{t=0}, \ldots, \hat{\mathbf{y}}_t) = \frac{\exp(\hat{\mathbf{y}}^\omega_t / T)}{\sum^\Omega_\omega \exp(\hat{\mathbf{y}}^\omega_t / T)}
\tag{5}
$$

where $T$ is the temperature value, and $P$ the probability of the output representation $\hat{\mathbf{y}}_{t+1}$ being the character $\omega$ given all previous outputs. The character sampling process was regulated by the temperature parameter $T$. When $T$ is set to a high value ($T \to \infty$), character probabilities tend to equalize across all characters. Conversely, as $T$ decreases towards 0, the highest likelihood predicted by $\hat{\mathbf{y}}_{t+1}$ approaches 1. In the context of DRAGONFLY applications, four distinct temperature values (0.2, 0.5, 0.8, 1.1) were investigated. A value of

$T = 0.5$ was found to strike the most favorable balance between novelty, diversity, the prediction of active compounds, and synthesizability, as indicated by the outcomes presented in Figs. S9–S10.

**Atom featurization**
*Small molecules*: The atomic properties of small-molecule ligands were encoded via the following embeddings: 10 atom types [H, C, N, O, F, P, S, Cl, Br, I], two ring types [True, False], two aromaticity types [True, False], and four hybridization types [$sp^3$, $sp^2$, sp, s].

*Proteins*: The protein binding site was defined by all protein atoms that are within a 5 Å radius to a ligand atom. The atomic properties of the respective protein binding sites were encoded using the following four features: (i) an embedding of the atom types using 22 different embeddings, (ii) an embedding of the combination of amino acid and atom types covering 225 different embeddigs, (iii) the distance to the closest atom of the bound small-molecule ligand, (iv) the calculated B factor, aiming to quantify protein flexibility and intrinsic disorder at the corresponding atom (Section S3).

*Bond types*: Edges were represented by inter-atomic distance in terms of Fourier features, using a sine- and cosine-based encoding for 3D graphs[82]. No edge features were used for 2D graphs. Edges were introduced between covalently bound atoms for the 2D graphs, and between all atoms within a 4 Å radius from each other for the 3D graphs.

**Hyperparameters**
The selected hyperparameters for the neural network led to a combined count of trainable parameters amounting to 6.94 million (3.49 million for the GTNN encoder and 3.45 million for the LSTM decoder) for the ligand-based design DRAGONFLY model. Similarly, the structure-based design DRAGONFLY model encompassed 7.01 million trainable parameters (3.56 million for the GTNN encoder and 3.45 million for the LSTM decoder).

**Scoring**
**Quantitative structure-activity relationship.** Kernel ridge regression (KRR) was employed to establish QSAR models based on descriptors and fingerprints. Kernel-based machine learning, rooted in the work of Krige[83], resides within the realm of supervised learning techniques and has found application across a spectrum of machine learning investigations[84–87]. The assessment of similarity between two molecules $i$ and $j$ was carried out utilizing the Laplacian Kernel (Eq. (6)):

$$
k(\mathbf{x}_i, \mathbf{x}_j) = \exp\left(-\frac{||\mathbf{x}_i - \mathbf{x}_j||_1}{\sigma}\right)
\tag{6}
$$

where $\mathbf{x}_i$ is the molecular descriptor or fingerprint of molecule $i$ and $\sigma$ is the length scale hyperparameter. Herein, $\sigma$ was set to 51.2, after screening $0.12^i$ for $i$ in range (1, 20). Three different molecular descriptors were applied in this study, namely, extended-connectivity fingerprints (ECFP, radius = 2, dimension = 512)[36], chemically advanced template search (CATS) with absolute feature frequencies[67,88], and ultrafast shape recognition with pharmacophoric constraints (USRCAT)[38]. Once the kernel matrix $\mathbf{K} = k(\mathbf{x}_i, \mathbf{x}_j)$ was calculated, the fitting coefficients $\boldsymbol{\alpha}$ were computed via the inverse of the kernel matrix $\mathbf{K}$ via Equation (7):

$$
\boldsymbol{\alpha} = (\mathbf{K} + \lambda\mathbf{I})^{-1}\mathbf{y}
\tag{7}
$$

where $\lambda$ denotes the regularization strength (herein, optimized to $10^{-7}$), $\mathbf{I}$ the identity matrix, and $\mathbf{y}$ the labels of the molecules (herein bioactivity to the investigated target). Given a labeled data set with $N$ molecule-label pairs $\{(\mathbf{x}_i, y_i)^N_{i=1}\}$, a function was obtained that maps the molecular descriptor of a novel molecule $\mathbf{x}_q$ to its predicted bioactivity

$\hat{y}_q$ via Equation (8):

$$\hat{y}_q(\mathbf{x}_q) = \sum_{i}^{N} \boldsymbol{\alpha}_i \cdot k(\mathbf{x}_i, \mathbf{x}_q) \qquad (8)$$

**Molecular novelty.** The novelty of the generated molecules was assessed through two distinct metrics: structural novelty score ($S_{ECFP}$) and scaffold novelty score ($S_{scaffold}$). The structural novelty score ($S_{ECFP}$) was established based on the Jaccard distance (1 minus Tanimoto similarity[89]) concerning the most similar molecule within the training data set using ECFP[36] descriptors. The Jaccard distance attains a value of 1 between two molecules when they possess no common structural attributes as identified by ECFP (bits within the ECFP vector). Conversely, it reaches a value of 0 when two distinct molecules share identical structural features (identical ECFP vectors). The scaffold novelty score ($S_{scaffold}$) gauges the novelty of both the atom scaffold (commonly referred to as the Murcko scaffold[90]) and the carbon scaffold (also known as the skeleton scaffold[91]) present in a generated molecule. Atom scaffolds were determined by considering the rings and branches of a specific template molecule. In this process, substituents were eliminated, while the identity of atoms and bonds remained unaltered (as detailed in SI2.4). Carbon scaffolds were identified by the carbon framework of a molecule, wherein all non-hydrogen atoms were transformed into carbon atoms and all bonds were replaced by single bonds (illustrated in Fig. S7). The scaffold novelty score was formulated by incorporating both atom and carbon scaffold scores. Each of these scores determined whether the corresponding scaffold was present in any molecule within the training set, as determined by Equations ((9)– (11)).

$$S_{atom} = \begin{cases} 0, & \text{if atom scaffold in training set} \\ 0.1, & \text{otherwise} \end{cases} \qquad (9)$$

$$S_{carbon} = \begin{cases} 0, & \text{if carbon scaffold in training set} \\ 0.1, & \text{otherwise} \end{cases} \qquad (10)$$

$$S_{scaffold} = S_{atom} + S_{carbon} \qquad (11)$$

Both structural and scaffold novelty contribute to the overall novelty score, i.e., Equation (12), ranging from 0 (for molecules very close to molecules the training set) to 1.2 (for molecules with no ECFP overlap with the training set and no shared scaffolds).

$$S_{novelty} = S_{ECFP} + S_{scaffold} \qquad (12)$$

**Molecular property analysis**
Molecular data sets were generated using a DRAGONFLY model, which was trained on a comprehensive data set excluding proteins and ligands associated with 20 specified targets. These targets are listed in Tables S2 and S3. For each target 2000 random molecules were selected. The physicochemical properties of these molecules were computed and subsequently used as input for the DRAGONFLY model. The properties of the generated molecules were visualized in a scatter plot (Fig. 2a) and summarized in Table 3. The scatter plot illustrates the relationship between the actual and predicted properties of the molecules. The mean absolute errors (MAEs) and Pearson correlation coefficients ($r$) were calculated to assess the predictive performance of the DRAGONFLY model. These statistical measures were derived by comparing the extracted properties of the generated molecules against the properties of the original data set.

**Drug-target interactome preprocessing**
The data necessary for constructing the drug-target graph, referred to as the "interactome," was sourced from two distinct databases: ChEMBL[28] (Version 29) and PDBBind[92] (Version 2020).

**Preprocessing ChEMBL data.** To acquire the necessary interactome data, the ChEMBL29 database[28] was queried. Similar to prior studies[93], this data extraction process was divided into two stages: In the initial step, a compilation of biological targets was obtained. Subsequently, compounds were extracted for which specific activities against any of these targets were annotated. Single-protein targets that possessed assay information for a minimum of 10 compounds with unique internal identifiers were retrieved from the ChEMBL database. A series of activity and annotation filters were then applied to these compounds. The molecules underwent neutralization, and any salts and solvents were eliminated. For compounds comprising multiple distinct fragments following this "washing" procedure, all but the fragment with the highest number of heavy atoms were discarded. Furthermore, molecules containing <3 or >100 heavy atoms, as well as radical species, were excluded from the data set. This procedure yielded a data set of 742 k unique SMILES-strings with annotated biologic affinity. Using a cut-off of a binding affinity of 200 nM, removing duplicates, a maximal SMILES-string length of 97 (using the longest SMILES-length from five randomized sampled SMILES-strings) for the ligand, and a minimum number of five ligands per target resulted in a drug-target graph consisting of 501 k unique binding affinities for 360 k unique ligands and 2989 unique target-IDs.

**Preprocessing PDBbind.** The PDBbind database (Version 2020) was obtained by downloading it from the link http://www.pdbbind.org.cn/download.php, which yielded a collective count of 19,443 protein-ligand structures. After filtering out structures annotated with "incomplete ligand structure", "covalent complex," or "incomplete ligand structure", a total of 19,000 entries remained. Additionally, a more refined filtering process was conducted, excluding structures with ligand molecular weights outside the range of 100–1200 g mol$^{-1}$ and binding affinities >10 $\mu$M. This filtration yielded a collection of 17,824 structures. This curated list of entries was then cross-referenced with the target-IDs present within the drug-target graph used for ligand-based design. This specific graph contained 501,000 unique binding affinities encompassing around 360,000 unique molecules and 2989 unique target-IDs. The outcome of this mapping effort revealed a total of 8351 distinct protein structures associated with 744 unique target-IDs. By refining the drug-target graph to exclusively include target-IDs with annotated PDB structures, the modified graph encompassed around 263,000 unique binding affinities spanning around 208,000 unique molecules and 744 unique target-IDs. The connection between PDB-IDs and target-IDs within ChEMBL was facilitated through UNIPROT-IDs, given that both databases provide UNIPROT-IDs for individual proteins.

Numerous drug targets exhibit multiple binding sites, including orthosteric sites and various allosteric sites[94]. Although such details were not present in the ChEMBL database, recognizing these distinct binding sites was deemed essential for effective drug-target interactome learning.molecules known for their allosteric modulation were extracted from the reference cited as Ref. 95. Subsequently, the drug-target graph underwent a modification whereby target-IDs encompassing both allosteric and orthosteric ligands were treated as distinct target-IDs.

**Chemical alphabet**
DRAGONFLY models underwent training using two distinct chemical alphabets: SMILES strings[3] and SELFIES[40]. To discern the distinct character types in both types of strings, 10 randomly generated SMILES strings were created for each molecule within the data set. For

SMILES strings, all observed characters surrounded by brackets ([]), as well as some frequently occurring functional groups (e.g., sulfoxide, nitro, ketone, nitrile) were encoded as a single token (SI5). In both string types, three supplementary characters were introduced to serve as markers for the beginning, end, and padding of the strings: $x$, $y$, and $z$ for SMILES-strings, and [\\X], [\\Y], and [\\Z] for SELFIES. Following this procedure, a SMILES-string alphabet $\Omega_{SMILES}$ was established, comprising a total of 57 characters. A SELFIES alphabet $\Omega_{SELFIES}$ was constructed, encompassing a total of 85 characters (as detailed in Table S1).

## Absolute free binding energy calculations

Molecules **1** and **2** as well as different ligands from ChEMBL with known PPARγ activity (ChEMBL IDs: ChEMBLl391987, ChEMBL241472, ChEMBL241299, ChEMBL213355, ChEMBL212591) were modeled into the PPARγ-aleglitazar crystal structure (PDB ID: 3G9E)[41]. The chosen reference molecules from the ChEMBL database were selected based on their structural similarity to compounds **1** and **2** (i.e., possessing (i) a carboxylic acid as head group, (ii) an alkyl or polyethylene glycol linker, and (iii) an aromatic tail), and their comparable binding affinity (i.e., $EC_{50}$ values $\leq 5\,\mu M$ and $\geq 100\,nM$). After structure preparation, ABFEP simulations were carried out with Schrödinger software (release 2023-4) using default settings and a simulation time of 5 ns for both complex and solvent[96]. The lowest calculated free energies were obtained for the co-crystallized ligand aleglitazar ($EC_{50} = 21\,nM$) and ChEMBL241472 ($EC_{50} = 140\,nM$) (Fig. S17).

## Cytotoxicity assay on HEK293T cells

HEK293T cells were seeded at the indicated number per well in DMEM-high glucose, complemented with glutamax, pen-strep, and 10% FBS, in a total of $40\,\mu l$ of medium. The cells were incubated overnight at 37 °C. Compounds were added to the cells at the indicated concentrations, resulting in a final Dimethylsulfoxid (DMSO) concentration of 0.2%. The compounds were incubated on the cells for either 16 h or 24 h. At the specified time point, the medium was carefully removed from the vessel, leaving only $2\,\mu l$ in the wells. Celltiter-glo (CTG) reagent (G7572, Promega) was prepared according to the manufacturer's instructions. Plates with cells were equilibrated at room temperature for 30 min. Subsequently, $25\,\mu l$ of CTG reagent was added to the cells. The plates were then shaken for 2 min and incubated for an additional 15 min at room temperature. Luminescence was read afterward with BG Pherastar.

## Biological characterization

Compounds **1**–**3** were characterized in a hybrid reporter gene assay for their agoniztic effect on human nuclear receptors PPARα/γ/δ, RXRα, FXRα, RARα in HEK293T cells. Compound **1** was tested in an isothermal titration calorimetry (ITC) assay to measure direct binding affinity to the ligand-binding domain of PPARγ. ADME properties were measured in standardized assays at Roche.

## Hybrid reporter gene assays.

PPAR activation was determined in uniform Gal4-hybrid reporter gene assays for the PPARα, PPARγ and PPARδ isoforms in HEK293T cells (German Collection of Microorganisms and Cell Culture GmbH, DSMZ) which were transiently transfected with pFR-Luc (Stratagene, La Jolla, CA, USA; reporter) and pRL-SV40 (Promega, Madison, WI, USA; internal control) and one pFA-CMV-hPPAR-LBD[97] clone, coding for the hinge region and ligand binding domain of the canonical isoform of human PPARα, PPARγ, PPARδ or respectively. Cells were cultured in Dulbecco's modified Eagle's medium (DMEM), high glucose supplemented with 10% fetal calf serum (FCS), sodium pyruvate (1 mM), penicillin (100 U · ml⁻¹), and streptomycin (100 μg · ml⁻¹) at 37 °C and 5% $CO_2$ and seeded in 96-well plates ($3 \times 10^4$ cells per well). After 24 h, medium was changed to Opti-MEM without supplements and cells were transiently

transfected using Lipofectamine LTX reagent (Invitrogen) according to the manufacturer's protocol. Five hours after transfection, cells were incubated with the test compounds in Opti-MEM supplemented with penicillin (100 U · ml⁻¹), streptomycin (100 μg · ml⁻¹) and 0.1% DMSO for 16 h before luciferase activity was measured using the Dual-Glo Luciferase Assay System (Promega) according to the manufacturer's protocol on a Tecan Spark luminometer (Tecan Deutschland GmbH, Germany). Firefly luminescence was divided by Renilla luminescence and multiplied by 1000 resulting in relative light units (RLU) to normalize for transfection efficiency and cell growth. Fold activation was obtained by dividing the mean RLU of a test compound by the mean RLU of the untreated control. All samples were tested in at least three biologically independent experiments in duplicates. For dose-response curve fitting and calculation of $EC_{50}$ values, the equation "[Agonist] versus response (variable slope−four parameters)" was used in GraphPad Prism (version 7.00, GraphPad Software, La Jolla, CA, USA) with fold activation data. The reference agonizts GW7647 (PPARα)[98,99], pioglitazone (PPARγ)[100,101] and L165,041 (PPARδ)[102,103] were used to validate the assays and to monitor assay performance. Nuclear receptor selectivity profiling was performed with corresponding pFA-CMV-hNR-LBD clones and suitable reference agonizts on RARα (pFA-CMV-hRARα-LBD[104], 1 μM tretinoin), LXRα (pFA-CMV-hLXRα-LBD[104], 1 μM TO901317) and RXRα (pFA-CMV-h RXRα-LBD[105], 1 μM Bexarotene).

## Isothermal Titration Calorimetry (ITC).

ITC experiments were conducted on an Affinity ITC instrument (TA Instruments, New Castle, DE) at 25 °C with a stirring rate of 75 rpm. PPARγ LBD protein (30 μM, prepared as described previously[106]) in buffer (20 mM Tris pH 7.5, 150 mM NaCl, 5% glycerol) containing 5% DMSO was titrated with the test compound (**1**) (100 μM in the same buffer containing 5% DMSO) in 21 injections ($1 \times 1\mu l$ and $20 \times 5\mu l$) with an injection interval of 120 s. The test compound was titrated into buffer, and the buffer was titrated to the PPARγ LBD proteins under otherwise identical conditions. The ITC results were analyzed using NanoAnalyze software (TA Instruments, New Castle, DE) with an independent binding model.

## Protein-ligand co-crystallization

The following construct was used for expression and co-crystallization. PPARγ (L204-Y477) (UniProt ID: P37231-2): MGSS-6His-SG-TEV-(L204-Y477). Molecular weight: 33465 Da. Large-scale expression of human PPARγ was conducted in E. coli BL-21 (DE3) cells (SI10). Subsequently co-crystals of PPARγ were grown using 6 mg · ml⁻¹ protein in buffer: 20 mM Tris-HCl pH 8.0, 1 mM TCEP, 0.5 mM EDTA and 1 mM design **1** mixed with equal amounts of reservoir: 0.1 M Tris-HCl pH 7.5 and 1.6 M ammonium sulfate (Fig. S14). The structure was determination and refinement yielding the elucidated co-crystal structure with a resolution of 1.85 Å as depicted in Fig. 5 (Table S13 and Fig. S15).

## Off-target screening

To test the specificity of compound **1** and **2**, both were subject to panel screen against 50 safety-relevant off-targets[107]. Both compounds have shown a clear profile not reaching ≥50% inhibition or binding at a concentration of 10 μM with the exception for PPARγ (Tables S9−S12).

## Chemical synthesis

Compounds **1**–**3** were synthesized starting from commercial building blocks. The synthesis and the full analytical characterization of the final compounds and intermediates are described in SI13.

## Co-crystallization

Compound **1** was co-crystalized with the ligand binding domain of human PPARγ (Leu²⁰⁴–Tyr⁴⁷⁷) (UniProt ID: P37231-2). The crystallographic structure is accessible from the Protein Data Bank[108] (PDB ID: 8PBO). Details about construct design, protein expression and

purification, crystallization, data collection, and structure determination and refinement can be found in SI10.

## Data availability

Source data is provided in Source_Data.zip and available on Figshare, https://doi.org/10.6084/m9.figshare.25234159, represented by https://doi.org/10.6084/m9.figshare.25234159[109]. The individual files in the ZIP file are named according to their location in the manuscript, for example, Figure_2_MolLogP.csv or Figure_6.pdb. Source data are provided with this paper.

## Code availability

A reference implementation of the DRAGONFLY method based on PyTorch[110] and PyTorch Geometric[111] is available at https://github.com/ETHmodlab/dragonfly_gen(rep. https://doi.org/10.5281/zenodo.10671327, https://zenodo.org/record/10671327)[112].

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

## Acknowledgements

Sarah Haller is thanked for technical support. Matthias Wittwer and Aynur Ekiciler are greatly acknowledged for generating ADME data. Karina M. Hugentobler is thanked for helpful scientific discussion. C.I. acknowledges support from the Scholarship Fund of the Swiss Chemical Industry. This research was supported by the Swiss National Science Foundation (SNSF, grant no. 205321_182176 and CRSII5_202245).

## Author contributions

Correspondence: Gisbert Schneider. K.A.: Conceptualization, data curation, formal analysis, investigation, methodology, initial software development, deep learning architecture design, model training and validation, visualization, writing—original draft. L.C.: Chemical synthesis, isolation, and characterization of compounds **1**–**3**, writing of the experimental procedure. C.I.: Conceptualization, formal analysis, investigation, methodology, software development, validation, visualization, writing—original draft. M.Hå.: Protein-ligand co-crystallization, writing—review and editing. D.F.: Protein-ligand co-crystallization, writing—review and editing M.Hi.: Methodology, software validation, visualization, writing—review and editing. D.F.N.: Conceptualization, methodology, writing—review and editing. M.I.: Methodology—validation of the ranking and scoring procedure. J.L.: Methodology—library creation with the RNN baseline model. C.C.G.S.: Methodology—validation of the ranking and scoring procedure. V.R.: Biological characterization, cytotoxicity assays, writing—review and editing. J.A.H.: Formal analysis, investigation, methodology, supervision, writing—review and editing. D.M.: Biological characterization, binding and functional assay, writing—review and editing. P.S.: Formal analysis, investigation, methodology, supervision, writing—review and editing. B.K.: ABFEP calculations, formal analysis, investigation, visualization, methodology, writing—review and editing. U.G.: Formal analysis, investigation, methodology, supervision, project administration, writing—original draft. G.S.: Formal analysis, investigation, methodology, supervision, funding acquisition, project administration, writing—original draft.

## Competing interests

G.S. and P.S. declare a potential financial and non-financial conflict of interest as co-founders of inSili.com LLC, Zurich, and in their role as scientific consultants to the pharmaceutical industry. D.F.N., V.R., U.G. and B.K. declare potential financial and non-financial conflict of interest

as full employees of F. Hoffmann-La Roche Ltd. M.Hå. and D.F. declare potential financial and non-financial conflict of interest as full employees of SARomics Biostructures AB. K.A., L.C., C.I., M.Hi., J.L., C.C.G.S., J.A.H. and D.M. declare no competing interest.
