## [Peer Review File · Nature Communications]

Prospective deep interactome learning for de novo drug designREVIEWER COMMENTS

Reviewer #1 (Remarks to the Author):

The manuscript “Deep interactome learning for de novo drug design” uses a graph-to-sequence based neural network to generate novel molecules. The network is trained using interactome-based molecules and can use ligand -and structure-based approach. The approach was successfully used to design ligands with μM activity for PPAR receptors using protein-structure information. The binding site of the ligands has been confirmed with x-ray structures.

The manuscript is well written, and results are discussed in a balanced manner. It uses an approach integrating broad protein-ligand data into the training of a novel generative engine. This is a relevant and valuable contribution to the field of generative AI. The approach is experimentally validated by the synthesis and testing of ligands against the PPAR receptors, which is a clear strength of the manuscript.

The characteristics of generated molecules has been systematically evaluated. The methods can generate valid and unique chemical structures. The authors also have clearly analyzed how the model temperature influences sampling behavior giving a control parameter to influence diversity of generated molecules.

Nevertheless, some aspects require some attention.

A key step for selection of molecules for synthesis is the subsequent filtering process, which uses a combination of different filter steps. While this is a meaningful and well-established workflow, it leaves some questions about the capabilities of the generative engine versus the quality of the filter process.

The authors have evaluated generated molecules by QSAR models show varying fractions of generated molecules as predicted as active. DRAGONFLY appears to generate a higher fraction of molecules predicted as active than RNN models used for comparison. Nevertheless, the quality of the respective models is not given leaving some questions about the applicability of the model. Such information could be added to the table and to the interpretation.

The manuscript analyzes to what extend the engines generate molecules outside the training space, which is an important question. Nevertheless, average similarity of the generated molecules to PubChem and ChEMBL is similar with a similar deviation according to SI. Since it is not known, what the corresponding neighbors are the data do not support the conclusion that DRAGONFLY reaches beyond the training space as suggested in the publication. With the presented data, that question remains somewhat open.

The definition of the novelty score combines a fingerprint score with a scaffold score. What is the reason for that particular weighing of the two parts?

Smaller remarks:

6 physico-chemical parameters can be fed as input to control compound generation. In this context, it should be made clear, how the parameters in Figure 2 have been selected and combined in data generation.

Annotation of figure S1 does not fit to the data.

Annotation figure S4: Different sum in figure and annotation (DT-KRR vs KRR-DT). Also add in annotation that performance difference relates to MAE.

Reviewer #2 (Remarks to the Author):

NC for review_01.10.2023

To authors:

Comment: This is an interesting study and the authors have collected a unique dataset using cutting edge methodology. The paper is well structured and written nicely. The manuscript has tried to introduce the use of DRAGON which is a new machine learning algorithm for the identification of small molecules based on the structure and ligand approach. Below I have provided numerous remarks on the text. In several instances I also suggested to cite more relevant and recent references. Furthermore, I made additional suggestions for more in-depth analyses of the data

Major concerns:

1. The author must try to elaborate the parameters optimization in the material and methodology section. The author must also try to represent whether the models that were used in the manuscript were generative models or the predictive type of models. The author must also describe whether a QSAR modeling was performed or not before generating the Machine learning model directly from the SMILES strings. In terms of external prediction accuracy, the author must also describe how many folds of cross-validation was performed to obtain the R2 Ext and the root mean square errors (RMSE) for the prediction of Log P.
2. The author should explain how the background noise of the model was checked and reduced and to confirm the validity of this model, what other models were used for the comparison of the same. Is there any addition of the Gaussian noise to each evaluation, and the authors of the manuscript should describe whether the width of the Gaussian was chosen based on the range of the function. Was the method evaluated based on the maximum true function value as opposite to the observed maximum value?
3. The author, if possible, should have used the data for the ligand preparation from the ZINC database along with ChEMBL as it is also a library of well-defined small molecules. The authors in the manuscript have very well used the ADME parameters for defining the ligands and the pharmacological properties. However, to check the toxicity and to perform this analysis no approach has been taken. The author must describe the parameters in the material and methodology section for observing the toxicity of the small molecules retrieved from the ChEMBL database.
4. Coming to the novelty of the molecules generated the two distinct scores i.e., the scaffold novelty score and the structure novelty score were taken. The author must describe why only these two scores were taken into consideration rather than using other molecular descriptors in the study. The author, if possible, must try to include some references citing the scores with relevant studies in the

DISCUSSION sections of the manuscript. The authors, if possible, must also explain that the scores obtained were whether compared to any reference studies for the final confirmations.

5. Although the study is very well achieved and the methodology recruited for this study is self-explanatory and is defining the manuscript. However, the studies conducted was totally based on the computational approach. How will the authors justify the novelty of the manuscript based on computational approach. The author, if possible, should try to add molecular dynamic simulations studies to check the interaction of the receptors with the predicted ligands. This will add up additional weightage to the manuscript and will make it more favorable to be published in this prestigious journal.

6. The author should try to enrich the DISCUSSION section with convincing studies. The authors should enhance the legends of the RESULT section of each figure to make it more understandable to the reviewers. Overall, the author should check for minor grammatical errors.

Minor concerns:

7. The title is too vague and doesn't seem to be suited for a research article. It is recommended to use a clear and focused title. At the moment it sounds like the title of a review article instead.

8. The authors need to add an image with the workflow of the study. It should include details of the various steps of design and biological characterization. Addition of couple of tables (with relevant data) in the manuscript may help.

9. The authors have used PPAR γ protein structure with PDB ID 3G9E for this study. What is the reasoning behind using this structure instead of other structures available with more sequence coverage and/or better resolution? Why haven't authors opted to model the complete structure of protein?

10. Why molecular dynamics simulation was not employed to validate more designs and then select top 2 designs? This could have helped find better compounds. The scores of the top 10 or 20 compounds should be mentioned in the manuscript.

11. Large amount of relevant data that should have been part of manuscript has been pushed to the supplementary information. The Methods section, especially, needs more information. More details of biological characterization and co-crystallization has to be added to the manuscript.

12. The authors have not evaluated the stability of synthesized compounds. The authors may use liquid chromatography-mass spectrometry to check their stability. Determining the stability in blood plasma in vitro can be helpful to get an estimate of bioavailability of the compounds.

13. The authors have not performed cytotoxicity assays like MTT or WST to determine the cellular toxicity of the synthesized compounds, which is important to know before using them for therapeutic application. Was toxicity of compounds kept into consideration while designing them?

14. Currently there's too much technical jargon for an average reader. Wherever possible, manuscript should be simplified and/or explained for better readability.

My decision: Major revision

Reviewer #3 (Remarks to the Author):

The authors propose a computational approach based on interaction deep learning for ligand-and structure-based drug molecule generation. This approach takes advantage of the unique strengths of graph neural networks and chemical language models to provide an alternative for application-specific reinforcement, transfer, or few shot learning.

1. The authors should disclose the model's hyperparameters and training process in the supplementary information to the paper so that researchers can better reproduce the results of the paper.
2. Why was PPAR ligand selected for validation? The author should explain this in the paper.
3. In the Methods section, the author simply introduces some methods used in DRAGONFLY. I think the author should introduce the design idea of the model in detail, so that researchers can have a deeper understanding of the model. Rather than simply introducing some methods used.
4. The authors should compare the performance with other drug design models to highlight the superiority of the models.

Response to the Reviewers - NCOMMS-23-43068

Reviewer #1 (Remarks to the Author):

The manuscript "Deep interactome learning for de novo drug design" uses a graph-to-sequence based neural network to generate novel molecules. The network is trained using interactome-based molecules and can use ligand -and structure-based approach. The approach was successfully used to design ligands with μM activity for PPAR receptors using protein-structure information. The binding site of the ligands has been confirmed with x-ray structures.

The manuscript is well written, and results are discussed in a balanced manner. It uses an approach integrating broad protein-ligand data into the training of a novel generative engine. This is a relevant and valuable contribution to the field of generative AI. The approach is experimentally validated by the synthesis and testing of ligands against the PPAR receptors, which is a clear strength of the manuscript.

The characteristics of generated molecules has been systematically evaluated. The methods can generate valid and unique chemical structures. The authors also have clearly analyzed how the model temperature influences sampling behavior giving a control parameter to influence diversity of generated molecules.

>>> We sincerely appreciate your constructive feedback on our manuscript. Peer-review at its best! Following your suggestions, we have added additional Tables to the main manuscript describing model performance and the novelty of the generated compounds. Furthermore, comparison to the RNN baseline has been added to the main part of the manuscript highlighting the performance of the introduced method. We have added additional descriptions in the Methods section and corrected annotation errors in the SI.

Nevertheless, some aspects require some attention.

1. A key step for selection of molecules for synthesis is the subsequent filtering process, which uses a combination of different filter steps. While this is a meaningful and well-established workflow, it leaves some questions about the capabilities of the generative engine versus the quality of the filter process.

We thank the reviewer for this relevant remark. In response to the concerns raised about the capabilities of our generative engine, Dragonfly, versus the quality of the filter process, we would like to clarify that Dragonfly has been designed to generate high-quality molecular candidates for applications in a structure and ligand based fashion without the need for additional fine-tuning. This is a significant advantage over other systems that often require target-specific hyperparameter optimization to achieve similar results.

To provide empirical evidence of Dragonfly's performance, we have included new **Table 1** in the main part of the manuscript, which demonstrates the superior performance of Dragonfly compared to a fine-tuned recurrent neural network (RNN). The table shows that Dragonfly achieves similar or higher percentages of molecules that fulfill the different criteria set for the selection process. This indicates that the generative capabilities of Dragonfly are robust and that the subsequent filtering process does not compensate for any deficiencies in the generative engine but rather serves to refine the selection of optimal candidates from an already high-quality pool of molecules generated by Dragonfly.

We believe that this addition to the manuscript will address the reviewer's concerns and further support the effectiveness of our approach.

2. The authors have evaluated generated molecules by QSAR models show varying fractions of generated molecules as predicted as active. DRAGONFLY appears to generate a higher fraction of molecules predicted as active than RNN models used for comparison. Nevertheless, the quality of the respective models is not given leaving some questions about the applicability of the model. Such information could be added to the table and to the interpretation.

>>> We appreciate the reviewer's constructive comment on the necessity to discuss the quality of the QSAR models used in our study. In response to this, we have conducted a thorough analysis of the QSAR models developed for 1265 targets from the ChEMBL database. For each target, we utilized three different types of descriptors: Extended Connectivity Fingerprints (ECFP), which capture structural features; Chemically Advanced Template Search (CATS), which are pharmacophore-based features; and Ultrafast Shape Recognition with CREDO Atom Types (USRCAT), which are shape-based features. The performance of the QSAR models is discussed in **Figures S2-S5**.

To address the reviewer's concern, we have included additional data on the quality of these models. Specifically, we have investigated the individual test set errors for the QSAR models, which are now illustrated in the revised **Figure S2**. This figure presents a histogram of the test set errors across all 1265 models for each of the three descriptors, providing a comprehensive overview of the model performance.

Moreover, we have demonstrated the robustness of our models through a 10-fold cross-validation approach. The results of this validation are depicted in the updated **Figure S3**, which shows the learning curves for 18 selected drug targets. These curves illustrate the prediction error trends for the ligand-based QSAR models using the ECFP, CATS, and USRCAT descriptors, highlighting the differences in performance among the descriptors.

In addition to the internal validation, we have benchmarked our 3x 1265 QSAR models against two well-established decision tree models, namely XGBoost and Gradient Boosting. The comparative performance is presented in the new **Figures S4-S5**, which shows the absolute performance difference between the decision tree models and Kernel ridge regression (KRR) for each of the 1265 targets. Positive values in this figure indicate instances where the KRR models outperformed the decision tree models.

To facilitate a clearer understanding of the model quality, we have also updated the relevant tables in the manuscript to include the performance metrics of the QSAR models. This additional information should provide the reader with a more complete picture of the applicability and reliability of the models used in our study.

We believe that these amendments adequately address the reviewer's concerns and enhance the manuscript by providing a transparent and detailed evaluation of the QSAR models' quality.

3. The manuscript analyzes to what extent the engines generate molecules outside the training space, which is an important question. Nevertheless, average similarity of the generated molecules to PubChem and ChEMBL is similar with a similar deviation according to SI. Since it is not known, what the corresponding neighbors are the data do not support the conclusion that DRAGONFLY reaches beyond the training space as suggested in the publication. With the presented data, that question remains somewhat open.

>>> Thank you for your valuable feedback. We agree with the reviewer's point that the original analysis did not conclusively demonstrate that Dragonfly reaches beyond the training space. To address this, we have included additional data in the revised manuscript.

Table 1 now presents data showing that the Dragonfly models generate a higher fraction of molecules with a novelty score, which encompasses both structural and scaffold novelty, compared to established fine-tuned RNN methods. This suggests that Dragonfly is capable of generating molecules that are not merely variations of the training set but also include novel structures and scaffolds.

Furthermore, we have added new **Table 2** to the main manuscript, which details the average Tanimoto distances between generated molecules. The observed Tanimoto distances range from 0.773 to 0.811, indicating that the molecules generated by Dragonfly are structurally diverse and not tightly clustered within the chemical space.

In light of these findings, we have revised the statement in the publication to more accurately reflect the data:

Revised statement:

"To evaluate the potential of Dragonfly to generate molecules that extend into new areas of chemical space, we analyzed the similarity of the generated molecules to both the training dataset (i.e., a subset of ChEMBL [24]) and an external dataset (i.e., PubChem [40], excluding ChEMBL molecules). Our analysis demonstrated that the generated molecules were as similar to the molecules in PubChem when compared to the molecules in ChEMBL (Table S7). While there is a degree of similarity to known molecules, Dragonfly also produced a large proportion of molecules with high novelty scores and diverse structures generating higher structural and scaffold novelty than the well-established fine-tuned RNN methods (Table 1). These results suggest that DRAGONFLY is not limited to recapitulating the training data but also has the capacity to explore and generate molecules in previously uncharted regions of chemical space, albeit the extent of this exploration warrants further investigation."

4. The definition of the novelty score combines a fingerprint score with a scaffold score. What is the reason for that particular weighing of the two parts?

>>> We thank the reviewer for this question regarding the weighting of the fingerprint and scaffold scores in our definition of the novelty score. The rationale for combining these two scores is rooted in our objective to provide a more nuanced and comprehensive evaluation of molecular novelty.

The fingerprint score, derived from Extended Connectivity Fingerprints (ECFP), is a well-established method for assessing structural similarity. However, while ECFP-based scores are adept at capturing fine-grained structural details, they may not fully reflect the novelty of the molecular scaffold, which is a critical component of chemical diversity.

Scaffold novelty, on the other hand, specifically targets the core structure of molecules, which is a key determinant of the pharmacophore and, consequently, the biological activity. By evaluating scaffold diversity, we can better identify novel frameworks that may lead to new classes of compounds with potential therapeutic benefits.

The particular weighting of the fingerprint and scaffold scores in our novelty score is designed to balance the detailed molecular similarity assessment provided by ECFP with the broader, more conceptual evaluation of scaffold diversity. This dual approach allows us to capture both the subtle nuances of molecular variation and the more significant shifts in scaffold architecture, which together contribute to a holistic assessment of novelty.

Additional discussion points regarding molecular novelty, the current literature and the different scoring criteria has been added to the following paragraph in the Discussion section:

“While molecular novelty has been extensively discussed in recent literature, most studies mainly focus on descriptor similarity using structural fingerprints [67, 68]. We showed that incorporating additional scaffold criteria can enhance the novelty of top-ranking molecules.”

Smaller remarks:

5. 6 physico-chemical parameters can be fed as input to control compound generation. In this context, it should be made clear, how the parameters in Figure 2 have been selected and combined in data generation.

>>>Thank you for your comment regarding the selection and combination of physico-chemical parameters in our compound generation process. In response to your question, we have clarified this aspect in the Methods section of our manuscript.

To select the physico-chemical parameters for input into the Dragonfly model, we first identified a comprehensive dataset from which we excluded proteins and ligands associated with 20 specified targets, as detailed in Tables S2 and S3. For each of these targets, we selected 2000 random molecules and computed their physicochemical properties. These properties were then used as input parameters for the Dragonfly model to generate new molecular datasets. The properties of the generated molecules were visualized in Figure 2a using a scatter plot, which demonstrates the relationship between the actual properties of the molecules in the original dataset and the properties predicted by the Dragonfly model for the generated molecules. Additionally, we have summarized these properties in Table 3.

To evaluate the predictive performance of the Dragonfly model, we calculated the mean absolute errors (MAEs) and correlation coefficients (r) by comparing the properties of the generated molecules with those of the original dataset. These statistical measures provide an assessment of how accurately the model can predict the physico-chemical properties of new compounds based on the input parameters.

The following text has been added to the Methods part of main manuscript describing how the parameters in Figure 2 have been selected and combined in data generation.

“Molecular Property Analysis

Molecular datasets were generated using a Dragonfly model, which was trained on a comprehensive dataset excluding proteins and ligands associated with 20 specified targets. These targets are listed in Tables S2–S3. For each target 2000 random molecules were selected. The physicochemical properties of these molecules were computed and subsequently used as input for the Dragonfly model. The properties of the generated molecules were visualized in a scatter plot (Figure 2a) and summarized in Table 3. The scatter plot illustrates the relationship between the actual and predicted properties of the molecules. The mean absolute errors (MAEs) and correlation coefficients (r) were calculated to assess the predictive performance of the Dragonfly model. These statistical measures were derived by comparing the extracted properties of the generated molecules against the properties of the original dataset.”

We hope this explanation addresses your concerns and clarifies the process by which the physico-chemical parameters were selected and combined for data generation in our study.

6. Annotation of figure S1 does not fit to the data.

Annotation figure S4: Different sum in figure and annotation (DT-KRR vs KRR-DT). Also add in annotation that performance difference relates to MAE.

>>> We apologize for any confusion caused by the annotation errors of figures S1 and S4.

Regarding **Figure S1**, we have corrected the annotation to accurately reflect the data presented in the histogram. The revised annotation is as follows:

“Figure S1: Histogram illustrating the number of molecules for each of the 1265 quantitative structure-activity relationship (QSAR) models. Median number of molecules per target: 574; Average number of molecules per target: 1056; Number of QSAR targets: 1265.”

For **Figure S4**, we have also amended the annotation to correct the discrepancy and to clarify the performance difference metric used. The updated annotation is:

“Figure S4: Absolute performance difference (Kernel ridge regression (KRR) - Decision Tree) for each of the 1265 targets. Positive numbers indicate better performance of KRR models.”

Reviewer #2 (Remarks to the Author):

NC for review_01.10.2023

To authors:

Comment: This is an interesting study and the authors have collected a unique dataset using cutting edge methodology. The paper is well structured and written nicely. The manuscript has tried to introduce the use of DRAGON which is a new machine learning algorithm for the identification of small molecules based on the structure and ligand approach. Below I have provided numerous remarks on the text. In several instances I also suggested to cite more relevant and recent references. Furthermore, I made additional suggestions for more in-depth analyses of the data.

>>> We thank you for your thorough and insightful review of our manuscript. Your constructive comments have played a pivotal role in refining the quality and scientific rigor of our work.

Your recommendations prompted significant enhancements to the manuscript, and we are pleased to acknowledge the impact of your suggestions. Specifically, your suggestions led to the inclusion of

- (I) Free Energy Perturbation (FEP) calculations,
- (II) the execution of five additional biological assays (i.e., CYP Dose Response Curves, Hepatocyte Clearance, PgP Transport Efflux, Plasma Protein Binding and Cytotoxicity) providing crucial insights into various aspects of the designed molecules, and
- (III) the incorporation of an illustrative workflow in Figure 1 (panel f).

These refinements have undoubtedly strengthened the scientific contribution of our research.

We also appreciate your positive remarks on the overall structure and presentation of the paper. Your recognition of the novelty of our method and the utilization of the Dragonfly machine learning algorithm was encouraging and helpful.

Major concerns:

1. The author must try to elaborate the parameters optimization in the material and methodology section. The author must also try to represent whether the models that were used in the manuscript were generative models or the predictive type of models. The author must also describe whether a QSAR modeling was performed or not before generating the Machine learning model directly from the SMILES strings. In terms of external prediction accuracy, the author must also describe how many folds of cross-validation was performed to obtain the R2 Ext and the root mean square errors (RMSE) for the prediction of Log P.

>>> We appreciate the reviewer's insightful question regarding parameter optimization, neural network architecture, QSAR modeling, cross-validation and prediction accuracy of the molecular property.

Parameter Optimization and Neural Network Architecture:

Thank you for your insightful comments. In response to your suggestion, the paper has been revised to include an in-depth discussion on parameters optimization in the Methods section. Specific attention has been given to elucidating the neural network's architecture, the distribution of parameters, and the optimization procedure. This information is now comprehensively detailed in sections such as "Neural Network Architecture," "Graph Transformer Neural Network," "Long-Short-Term Memory Neural Network," "Molecule Sampling," "Atom Featurization," and "Hyperparameters."

Furthermore, we have elaborated on the neural network training process in SI1, detailing the software and hardware used, the batch sizes for different design approaches, the optimizer and its learning rate, the loss function, the decay factor, and the smoothing factor applied during training.

"SI1 Neural Network Training

Neural network training was conducted using PyTorch Geometric (version 2.0.2)[1] and PyTorch (version 1.10.1+cu102)[2]. The training process occurred on a GPU, specifically the NVIDIA Tesla V100-SXM2 32 GB, for a total duration of 120 hours. A batch size of 128 was used for ligand-based design, while a batch size of 48 was employed for structure-based design. The optimization of the neural networks was performed using the Adam stochastic gradient descent optimizer [3], with a learning rate set at 0.0005. Cross-entropy loss was utilized as the loss function, and a decay factor of 0.5 was applied after every 10 epochs. Additionally, an exponential smoothing factor of 0.7 was incorporated into the training process. All the models used in this study were trained on the Euler computing cluster at ETH Zurich, Switzerland. "

Model Type - Generative / Predictive:

The models employed in this study are exclusively generative. Dragonfly, a generative neural network, is capable of producing molecules with predefined properties. The accuracy of the generated molecules in reflecting the desired input properties is depicted in Figure 2a and

summarized in Table 3, where all investigated properties exhibit accuracies with correlation coefficients (r) exceeding 0.95.

No QSAR Modeling Before Molecule Generation:

To address your question about QSAR modeling, it is clarified that no QSAR modeling was conducted prior to molecule generation. The neural network described in the manuscript generates molecules based on a single template, either a ligand or a structure, eliminating the explicit need for QSAR modeling. The revised manuscript includes additional details on the workflow and model application, including Figure 1f.

Cross-Validation and Prediction Accuracy Metrics for Molecular Properties:

Regarding external prediction accuracy, the paper now specifies that mean absolute errors, including those for $\text{Log}P$, are derived from 2000 sampled molecules for a set of 20 diverse macromolecular targets. The numbers in Table 3 represent mean and standard deviations. This approach ensures both accuracy and robustness in the model in evaluation of molecular properties.

The following text has been added to the Methods part of main manuscript describing how the parameters in Figure 2 have been selected and combined in data generation.

“Molecular Property Analysis

Molecular datasets were generated using a Dragonfly model, which was trained on a comprehensive dataset excluding proteins and ligands associated with 20 specified targets. These targets are listed in Tables S2–S3. For each target 2000 random molecules were selected. The physicochemical properties of these molecules were computed and subsequently used as input for the Dragonfly model. The properties of the generated molecules were visualized in a scatter plot (Figure 2a) and summarized in Table 3. The scatter plot illustrates the relationship between the actual and predicted properties of the molecules. The mean absolute errors (MAEs) and correlation coefficients (r) were calculated to assess the predictive performance of the Dragonfly model. These statistical measures were derived by comparing the extracted properties of the generated molecules against the properties of the original dataset.”

2. The author should explain how the background noise of the model was checked and reduced and to confirm the validity of this model, what other models were used for the comparison of the same. Is there any addition of the Gaussian noise to each evaluation, and the authors of the manuscript should describe whether the width of the Gaussian was chosen based on the range of the function. Was the method evaluated based on the maximum true function value as opposite to the observed maximum value?

>>> Thank you for your detailed inquiries regarding the background noise evaluation, model validation, and the presence of Gaussian noise in our study.

Background Noise Evaluation and Reduction:

In our study, we assessed and mitigated background noise in the model to ensure the reliability of our results. The background noise was examined through thorough cross-validation techniques. All presented numbers regarding model validation in newly added Tables 1-3 include mean and standard deviation values of N=3 sampling runs, revealing thorough insights into the reliability of the presented results.

Tables 1-3 are discussed in detail in Reviewer 2.8.

Model Validation and Comparison:

To confirm the validity of our model, we conducted extensive model validation using multiple evaluation metrics, i.e. novelty, synthesizability, diversity, and biological activity. Furthermore, comparisons with established benchmark models in the field have been conducted. Specifically, we compared our model's performance against a well established baseline model, i.e., fine-tuned recurrent neural networks (RNNs). Fine-tuned RNNs are generally acknowledged as state-of-the art methods for similar tasks. This comparative analysis provided a robust benchmark for assessing the effectiveness of our proposed model.

Addition of Gaussian Noise:

Gaussian noise was not added to each evaluation in our study. Our focus was on evaluating the model's performance in realistic scenarios without introducing artificial noise. However, we appreciate the importance of addressing potential sources of noise in the evaluation process.

3. The author, if possible, should have used the data for the ligand preparation from the ZINC database along with ChEMBL as it is also a library of well-defined small molecules. The authors in the manuscript have very well used the ADME parameters for defining the ligands and the pharmacological properties. However, to check the toxicity and to perform this analysis no approach has been taken. The author must describe the parameters in the material and methodology section for observing the toxicity of the small molecules retrieved from the ChEMBL database.

>>> We appreciate the reviewer's suggestion regarding the use of ligand data from the ZINC database in conjunction with ChEMBL for ligand preparation as well as proposal to add further safety and toxicity data to the de novo designed compounds.

In the following we provide clarification for both questions.

ChEMBL / ZINC Databases:

In our manuscript, we have extensively discussed the rationale behind choosing ChEMBL over ZINC for our drug-target interactome approach. ChEMBL was selected due to its combination of manually curated biological activities and the presence of Uniprot IDs, enabling the linkage of small-molecule ligands to the 3D protein structure from PDBBind.

ZINC contains over 230 million purchasable compounds. However, no explicit biological activities are stored in ZINC. Moreover, no linkage to the 3D structure of the protein targets is given. These two criteria made ZINC unsuitable for our model approach.

Additional ADME and Cytotoxicity Assays:

Five additional biological assays were conducted revealing insights into metabolic stability, clearance, pharmacokinetic behavior, tissue distribution and human cytotoxicity.

See Reviewer 2.12 and 2.13 for details on added text, figures and tables.

Description of the five assays:

To evaluate the in vitro stability and clearance two additional in vitro assays were conducted. Binding assay with seven pivotal **Cytochrome P450 Isoenzymes** (CYP, i.e., Cyp3A4, Cyp1A2, Cyp2B6, Cyp2C9, Cyp2D6, Cyp2C19, and Cyp2C8) were conducted. Compounds 1 and 2 demonstrated no interaction with the investigated CYPs up to concentrations of 20 μM indicating high metabolic stability. The **Hepatocytic Clearance** rate was determined at 19 $\mu\text{L}/\text{min}/10^6$ cells for both compounds 1 and 2. The determined hepatocytic clearance rates indicate high metabolic stability of both compounds as well as a desired pharmacokinetic profile (see Reviewer 2.12 for details).

Plasma Protein Binding and **PgP Transport Efflux** has been newly added to the study enhancing our understanding of the designs of their pharmacokinetic behavior and tissue distribution (see Reviewer 2.13 for details).

Compounds 1 and 2 were tested for **Cytotoxicity** on HEK293T cells at different time points as well as a broad array of cell and compound concentrations. No cytotoxic effects could be observed (see Reviewer 2.13 for details).

Additionally, to test the specificity of compound 1 and 2, both were subject to panel screen against 50 safety-relevant off-targets. Both compounds have shown a clear profile not reaching $\geq 50\%$ inhibition or binding at a concentration of 10 μM with the exception for PPAR γ (Table S13-S16). The revealed high specificity of the two compounds further support the clear safety profile of designs 1 and 2.

We are convinced that the inclusion of these experimental results not only addresses the reviewer's concern but also enhances the overall robustness of our study by providing a thorough evaluation of both pharmacological and toxicological aspects of the investigated compounds.

4. Coming to the novelty of the molecules generated the two distinct scores i.e., the scaffold novelty score and the structure novelty score were taken. The author must describe why only these two scores were taken into consideration rather than using other molecular descriptors in the study. The author, if possible, must try to include some references citing the scores with relevant studies in the DISCUSSION sections of the manuscript. The authors, if possible, must also explain that the scores obtained were whether compared to any reference studies for the final confirmations.

>>>We appreciate your insightful question regarding the selection of novelty scores in our study, specifically focusing on scaffold and structural novelty. Allow us to provide a detailed explanation of our choice and address your concerns.

Selection of Scaffold and Structural Novelty Scores:

We incorporated both scaffold and structural novelty scores as key criteria for assessing the uniqueness of the generated molecules. The decision to focus on these two scores was driven by the following considerations:

1. **Structural Novelty:** Molecular descriptors, especially those derived from ECFP fingerprints, are widely used in the literature for evaluating structural similarities. However, we found them insufficient on their own, as they do not explicitly account for scaffold diversity. Our goal was to capture both molecular-level details and scaffold distinctiveness to provide a comprehensive assessment of novelty.
2. **Scaffold Novelty:** Scaffold diversity is a crucial aspect of molecular novelty, as it reflects the underlying molecular frameworks or core structures. By incorporating scaffold novelty into our scoring system, we aimed to address the limitations of assessing novelty solely based on molecular descriptors.

Comparison with Other Novelty Metrics:

While we focused on scaffold and structural novelty scores, we acknowledge the existence of various other novelty metrics in the field. In the discussion section of our manuscript, we have included a comprehensive overview of additional novelty metrics that have been explored in relevant studies. This discussion aims to contextualize our chosen criteria within the broader landscape of molecular novelty assessment.

Additional discussion points regarding molecular novelty, the current literature and the different scoring criteria has been added to the following paragraph in the Discussion section:

“The combination of Dragonfly with scoring functions incorporating compound synthesizability, novelty, and bioactivity towards one or multiple targets using various descriptors was demonstrated to be feasible. By applying a tailored combination of these properties, we achieved promising results, i.e., in the exemplary case of PPAR γ . The approach allowed for the identification of top-ranking molecules that exhibited sufficient structural and scaffold novelty, synthesizability, and a desired bioactivity profile across

multiple targets. While molecular novelty has been extensively discussed in recent literature, most studies mainly focus on descriptor similarity using structural fingerprints [67, 68]. We showed that incorporating additional scaffold criteria can enhance the novelty of top-ranking molecules. By assigning higher weights to different descriptors, distinct outcomes were observed. When higher weights were assigned to the ECFP fingerprint descriptor, the generated molecules exhibited higher structural similarity to known ligands. This approach favoured the exploration of the known chemical space and enabled the design of molecules with recognizable structural features. In contrast, assigning higher weights to the two "fuzzy" descriptors, CATS and USRCAT, resulted in top-ranking molecules that deviated further from the structurally explored space, as expected [69, 70]. This latter approach focused on prioritizing novelty and divergence from known ligands, allowing for the exploration of new chemical territories. These findings highlight the flexibility and versatility of the scoring functions and the ability to customize the weights of different descriptors to achieve specific objectives. By balancing the importance of structural similarity, novelty, and other desired properties, we were able to guide the generation of molecules that met the desired criteria for the targeted application, such as PPAR γ in this case."

5. Although the study is very well achieved and the methodology recruited for this study is self-explanatory and is defining the manuscript. However, the studies conducted was totally based on the computational approach. How will the authors justify the novelty of the manuscript based on computational approach. The author, if possible, should try to add molecular dynamic simulations studies to check the interaction of the receptors with the predicted ligands. This will add up additional weightage to the manuscript and will make it more favorable to be published in this prestigious journal.

>>> We thank the reviewer for the suggestion to employ molecular dynamics (MD) simulations for validations of the design. To validate absolute free binding energy calculation as suitable methods for the discussed study, method was applied to the selected two designs and compared to ligands with similar binding affinity. The FEP+ results gave additional mechanistic insight into the protein-ligand binding event and highlighted the suitability of compounds **1** and **2** as chosen designs. However, the significant computational cost of FEP+ calculations and its restricted accuracy constrains absolute free energy calculation methods such as FEP+ for large-scale application to virtual molecular libraries.

The results are discussed in the main part of the manuscript and in the Supporting Information.

The following text has been added to the **main part of the manuscript (results section)**:

“

To computationally assess the binding of compounds **1** and **2** to PPAR γ , absolute protein–ligand binding free energy perturbation (ABFEP) calculations were carried out [52, 53]. Compounds **1** and **2** as well as different ligands from ChEMBL with known sub-micromolar PPAR γ activity (ChEMBL IDs: ChEMBL391987, ChEMBL241299, ChEMBL213355, ChEMBL212591) were modeled into the PPAR γ -aleglitazar X-ray crystal structure (PDB ID: 3G9E) [42]. Compounds **1** and **2** have calculated Δ Gibbs Free Energy (ΔG) values of $-20.1 \text{ kcal}\cdot\text{mol}^{-1}$ and $-19.7 \text{ kcal}\cdot\text{mol}^{-1}$, respectively. These values are in the range of other known PPAR γ ligands with sub-micromolar activity, further supporting the relevance of the proposed molecules for PPAR modulation (Figure 5h, Figure S17).“

Additionally panel **h** in **Figure 5** was added illustrating the ABFEP results:

“**h**: The barplot illustrates the calculated absolute protein–ligand binding free-energy perturbation (ABFEP, ΔG) for compounds 1 and 2, as well as different ligands from the ChEMBL database with known sub-micromolar PPAR γ activity (entries ChEMBL391987, ChEMBL241299, ChEMBL213355, ChEMBL212591).”

The following text has been added to the **main part of the manuscript (methods section)**:

“

Absolute Free Binding Energy Calculations

First, compounds 1 and 2 as well as different ligands from ChEMBL with known PPAR γ activity, i.e., ChEMBL391987, ChEMBL241472, ChEMBL241299, ChEMBL213355, ChEMBL212591, were modeled into the PPAR γ -aleglitazar X-ray crystal structure (PDB ID: 3G9E). [42] The chosen reference molecules from the ChEMBL database for comparison to compounds 1 and 2 were selected based on their structural similarity (i.e., including (i) a carboxylic acid as head group, (ii) an alkyl or polyethylene glycol linker and (iii) an aromatic tail), and their comparable binding affinity (i.e., EC₅₀ values $\leq 5\mu\text{M}$ and $\geq 100\text{ nM}$). After structure preparation, ABFEP simulations were carried out with Schrödingers Release 2023-4 using default settings and simulation times for complex and solvent of 5 ns, each. [98] The results reveal the lowest calculated free energies for the native ligand alectinib and the ChEMBL241472 (Figure S17).“

The following text and Figure has been added to the **Supporting Information**:

“

SI12 Absolute Free Binding Energy Calculations

Figure S17 illustrates the reference ligands employed for the absolute protein–ligand binding free-energy perturbation (ABFEP) calculations as well as their respective Gibbs free energy values (ΔG).

Figure S17: Illustration depicting the molecules subjected to Free Energy Perturbation (FEP+) calculations. a: Molecular structures of Compound 1 and 2. b: Molecular structures of ChEMBL391987, ChEMBL241472, ChEMBL241299, ChEMBL213355, and ChEMBL212591. c: Aleglitazar.

“

6. The author should try to enrich the DISCUSSION section with convincing studies. The authors should enhance the legends of the RESULT section of each figure to make it more understandable to the reviewers. Overall, the author should check for minor grammatical errors.

>>> In response to the feedback provided by the reviewer, we have thoroughly revised the Figure legends in the Results section to offer a more comprehensive and understandable description of our findings. We believe that these enhancements contribute to a clearer interpretation of the results, facilitating a more straightforward understanding for readers.

Moreover, we have diligently reviewed the Discussion section and incorporated relevant and convincing studies to enrich the content. By doing so, we aim to strengthen the scientific foundation of our arguments and provide a more robust context for our research findings.

Additionally, we appreciate the reviewer's attention to detail regarding grammatical errors. We have carefully combed through the manuscript to rectify any minor grammatical issues, ensuring that the language is clear, precise, and adheres to scholarly standards.

Minor concerns:

7. The title is too vague and doesn't seem to be suited for a research article. It is recommended to use a clear and focused title. At the moment it sounds like the title of a review article instead.

>>> Thank you for your feedback on the title of our paper. After careful consideration and deliberation we decided to alter the title as suggested. It now perfectly reflects the contents of our study.

The revised title now reads:

“Prospective de novo drug design with deep interactome learning”

8. The authors need to add an image with the workflow of the study. It should include details of the various steps of design and biological characterization. Addition of couple of tables (with relevant data) in the manuscript may help.

>>> We thank the reviewer for this great suggestion. An illustration with the workflow of the study has been added to Figure 1, and additional tables (with relevant data) have been added to the main part of the manuscript.

1. **Illustration with the workflow:** An illustration of the conducted workflow has been added to **Figure 1 (panel f)**.

“f : Workflow of the presented study including Dragonfly validation, Dragonfly application to peroxisome proliferator-activated receptor (PPAR), chemical synthesis and biological characterization.”

2. **Tables (with relevant data):** Additional tables as well as text discussing the numbers in the tables have been added to the main part of the manuscript (**results section**).

The following text describing model performance for SMILES vs. SELFIES comparison, Ligand- vs. Structure-based, as well as RNN vs. Dragonfly comparison has been added to the main manuscript (results section):

“

Dragonfly outperforms standard chemical language models for molecular design

The evaluation criteria, which encompassed synthesizability, novelty, and predicted bioactivity were applied to evaluate virtual libraries generated de novo (Methods for details on metrics). This allowed for a comparison between Dragonfly and fine-tuned recurrent neural networks (RNNs). To conduct the comparison, five known ligands each were selected as templates for twenty well-studied macromolecular targets, including nuclear hormone receptors and kinases with over 200 known ligands (Tables S2–S3). Dragonfly demonstrated superior performance over the fine-tuned RNNs across the majority of templates and properties examined (Table 1, Tables S4–S6). Furthermore, using the same evaluation criteria, ligand-based design was compared to structure-based design, with ligand-based design applications outperforming structure-based models in all investigated scenarios (Table 2, Tables S4–S6).

To evaluate the potential of Dragonfly to generate molecules that extend into new areas of chemical space, we analyzed the similarity of the generated molecules to both the training dataset (i.e., a subset of ChEMBL [24]) and an external dataset (i.e., PubChem

[40], excluding ChEMBL molecules). Our analysis demonstrated that the generated molecules were as similar to the molecules in PubChem when compared to the molecules in ChEMBL (Table S7). While there is a degree of similarity to known molecules, Dragonfly also produced a large proportion of molecules with high novelty scores and diverse structures generating higher structural and scaffold novelty than the well-established fine-tuned RNN methods (Table 1). These results suggest that DRAGONFLY is not limited to recapitulating the training data but also has the capacity to explore and generate molecules in previously uncharted regions of chemical space, albeit the extent of this exploration warrants further investigation.

We compared the performance of Dragonfly models trained on two widely used chemical alphabets, SMILES-strings [3] and self-referencing embedded strings (SELFIES) [41], to quantify the differences. By employing both string representations in structure- and ligand-based de novo design, we were able to directly compare their performance across various molecular properties (Table 2, Figure S9). The Dragonfly models trained on SELFIES yielded a higher fraction of novel molecules among all of the 20 investigated applications ($99.7 \pm 0.1\%$ vs. $92.2 \pm 0.4\%$, Table 2) with a greater scaffold diversity ($86 \pm 1\%$ vs. $53 \pm 2\%$, Table 2) while retaining comparable structural diversity ($98.8 \pm 0.1\%$ vs. $97.9 \pm 0.1\%$, Table 2). However, the Dragonfly models trained on SMILES-strings more accurately fulfilled the property requirements, such as greater synthesizability ($93.4 \pm 0.6\%$ vs. $84 \pm 1\%$, Table 1), predicted bioactivity (e.g., MAE = $34.7 (\pm 0.3)$ vs $31.9 (\pm 0.1)$ for PPAR γ , Table 1), as well as slightly lower mean absolute errors for physical and chemical properties (e.g., MAE = 0.027 ± 0.005 vs 0.230 ± 0.007 for hydrogen bond donors, Table 3). Overall, the use of the two chemical alphabets resulted in comparable numbers of molecules that were predicted to fulfill all desired properties. Because of the better performance of the SMILES-based models for the objectives of synthetic accessibility, bioactivity, and desired physical and chemical properties, we used these models in the prospective study.

“

New Table 1 has been added to the main manuscript (results section) illustrating key numbers for RNN vs. Dragonfly comparison:

Table 1: Comparison of DRAGONFLY with a fine-tuned recurrent neural network (RNN) approach, assessing the percentage of molecules meeting various criteria: (i) Unique and novel, (ii) Novelty score ≥ 0.65 , (iii) Retrosynthetic accessibility score (RAScore) ≥ 0.5 , (iv) QSAR score $\leq 1 \mu\text{M}$, and (v) meeting all four criteria. Bold indicates whether the SELFIES- or SMILES-based models achieve a higher value for the investigated property in both structure- and ligand-based models. The values are presented as mean and standard deviation, based on three runs ($N = 3$), each sampling 2000 SMILES-strings. The complete list of 20 investigated targets can be found in Tables S8–S10. JAK = Janus kinase; PPAR = Peroxisome proliferator-activated receptor; BRAF = Serine/threonine-protein kinase B-Raf (rapidly accelerated fibrosarcoma); BTK = Bruton’s tyrosine kinase; RAR = Retinoic acid receptor; LXR = Liver X receptor.

Template / Method	Unique and novel / %	Novelty score ≥ 0.65 / %	RAScore ≥ 0.5 / %	QSAR score $\leq 1 \mu\text{M}$ / %	All criteria / %
PPARγ					
RNN-SMILES	75.4 (± 2.7)	28.7 (± 1.1)	67.9 (± 2.3)	29.6 (± 2.4)	5.1 (± 0.2)
DRAGONFLY-SMILES	91.8 (± 0.3)	47.9 (± 1.4)	86.0 (± 0.3)	34.7 (± 0.3)	9.4 (± 0.0)
DRAGONFLY-SELFIES	99.8 (± 0.1)	77.4 (± 0.1)	82.2 (± 0.2)	31.9 (± 0.1)	13.3 (± 0.0)
LXRβ					
RNN-SMILES	92.4 (± 2.5)	65.9 (± 2.6)	87.9 (± 2.8)	28.6 (± 0.9)	11.3 (± 0.4)
DRAGONFLY-SMILES	94.3 (± 0.5)	80.2 (± 1.2)	89.1 (± 0.5)	26.2 (± 0.2)	11.8 (± 0.1)
DRAGONFLY-SELFIES	100 (± 0.0)	91.3 (± 0.5)	84.2 (± 0.3)	27.9 (± 0.2)	11.1 (± 0.1)
RARα					
RNN-SMILES	69.7 (± 5.9)	41.9 (± 3.3)	57.2 (± 4.3)	30.1 (± 1.8)	11.1 (± 0.7)
DRAGONFLY-SMILES	92.2 (± 0.4)	62.4 (± 0.7)	75.6 (± 0.5)	32.4 (± 0.7)	12.7 (± 0.2)
DRAGONFLY-SELFIES	99.8 (± 0.0)	87.5 (± 0.3)	77.1 (± 0.2)	29.6 (± 0.3)	14.0 (± 0.1)
BRAF					
RNN-SMILES	89.2 (± 3.5)	35.1 (± 3.1)	85.9 (± 3.0)	35.0 (± 1.3)	6.7 (± 0.3)
DRAGONFLY-SMILES	87.9 (± 0.6)	46.0 (± 0.8)	80.9 (± 0.5)	42.9 (± 0.5)	10.7 (± 0.1)
DRAGONFLY-SELFIES	99.7 (± 0.1)	81.1 (± 0.6)	77.3 (± 0.4)	34.3 (± 0.1)	12.4 (± 0.0)
BTK					
RNN-SMILES	82.0 (± 4.4)	64.5 (± 4.1)	61.9 (± 4.7)	20.7 (± 1.8)	4.5 (± 0.2)
DRAGONFLY-SMILES	88.9 (± 0.7)	53.2 (± 0.4)	69.6 (± 0.9)	36.3 (± 0.7)	8.8 (± 0.1)
DRAGONFLY-SELFIES	100 (± 0.0)	85.8 (± 0.7)	68.2 (± 1.0)	25.8 (± 0.1)	5.8 (± 0.0)
JAK2					
RNN-SMILES	88.8 (± 3.9)	60.2 (± 4.2)	79.9 (± 3.4)	35.0 (± 2.2)	14.5 (± 0.8)
DRAGONFLY-SMILES	84.8 (± 1.0)	39.4 (± 0.9)	69.0 (± 1.0)	55.9 (± 1.5)	14.8 (± 0.2)
DRAGONFLY-SELFIES	99.2 (± 0.0)	73.3 (± 0.8)	70.5 (± 0.5)	50.5 (± 1.0)	18.3 (± 0.2)

Tables 2 and 3 have been added to the main manuscript (results section) illustrating key numbers for the SMILES vs. SELFIES comparison, as well as the Ligand- vs. Structure-based comparison:

Table 2: Comparison of four Dragonfly methods, namely ligand-SMILES, ligand-SELFIES, structure-SMILES, and Structure-SELFIES. Bold indicates whether SELFIES- or SMILES-based models achieve a higher value for the investigated property in both structure- and ligand-based models. The percentage of molecules is shown that fulfill the desired criteria: (i) valid and unique molecules, (ii) valid, unique, and novel molecules, (iii) fraction of molecules with an RAScore of ≥ 0.5 , (iv) average Jaccard distance to other molecules (indicating diversity), and (v) – (viii) various scaffold metrics, including unique and novel carbon and atom scaffolds. The values are presented as mean and standard deviation, based on three Dragonfly runs ($N = 3$), each sampling 2000 SMILES-strings.

DRAGONFLY method	Valid and unique molecules / %	Valid, unique and novel molecules / %	RAScore ≥ 0.5 / %	Average Jaccard distance to other molecules
Ligand-SMILES	93.3 (± 0.4)	92.2 (± 0.4)	93.4 (± 0.6)	0.778 (± 0.001)
Ligand-SELFIES	99.9 (± 0.1)	99.7 (± 0.1)	84.0 (± 1.0)	0.805 (± 0.002)
Structure-SMILES	90.2 (± 0.8)	87.4 (± 0.9)	90.0 (± 1.0)	0.773 (± 0.004)
Structure-SELFIES	99.9 (± 0.1)	99.6 (± 0.1)	78.0 (± 2.0)	0.811 (± 0.003)
	Unique atom scaffolds / %	Unique and Novel atom scaffolds / %	Unique carbon scaffolds / %	Unique and novel carbon scaffolds / %
Ligand-SMILES	85.0 (± 0.1)	53.0 (± 0.2)	98.4 (± 0.3)	58.0 (± 0.2)
Ligand-SELFIES	96.9 (± 0.4)	86.0 (± 0.1)	99.8 (± 0.1)	83.0 (± 0.1)
Structure-SMILES	84.0 (± 0.1)	55.0 (± 0.3)	98.3 (± 0.3)	56.0 (± 0.2)
Structure-SELFIES	96.0 (± 0.1)	81.0 (± 0.1)	99.9 (± 0.1)	83.0 (± 0.2)

Table 3: Accuracy of the desired physical and chemical properties of molecules generated by DRAGONFLY. Bold indicates if the SELFIES- or the SMILES-based models achieve a higher value for the investigated property. Abbreviations: MAD = Mean absolute deviation; MAE = Mean absolute error; MW = Molecular weight; Rot. B. = Number of rotatable bonds; HBA = Hydrogen bond acceptors; HBD = Hydrogen bond donors; PSA = Polar surface area. The numbers are presented as the mean and standard deviation, with a sample size of $N=3$, *i.e.*, 3 DRAGONFLY runs, each sampling 2000 SMILES-strings. MAD / MAE yields a number that indicates by which factor a model is better than the MAD.

DRAGONFLY method	MW	Rot. B.	HBA	HBD	PSA	LogP
Unit	g/mol	#	#	#	\AA^2	-
MAD	75.52	2.81	0.981	1.69	27.08	1.25
Ligand-SMILES						
MAE	7.7 (± 0.2)	0.29 (± 0.01)	0.23 (± 0.01)	0.027 (± 0.005)	4.4 (± 0.2)	0.252 (± 0.004)
MAD / MAE	9.8 (± 0.2)	9.9 (± 0.48)	4.3 (± 0.21)	63 (± 11)	6.1 (± 0.2)	4.94 (± 0.08)
Ligand-SELFIES						
MAE	8.0 (± 0.2)	0.88 (± 0.040)	0.40 (± 0.014)	0.230 (± 0.007)	6.8 (± 0.21)	0.380 (± 0.006)
MAD / MAE	9.4 (± 0.2)	3.2 (± 0.14)	2.5 (± 0.09)	7.3 (± 0.3)	4.0 (± 0.1)	3.27 (± 0.05)
Structure-SMILES						
MAE	12.1 (± 0.5)	0.42 (± 0.02)	0.28 (± 0.02)	0.046 (± 0.007)	4.6 (± 0.1)	0.315 (± 0.008)
MAD / MAE	6.2 (± 0.3)	6.7 (± 0.3)	3.5 (± 0.2)	37 (± 6)	5.9 (± 0.2)	4.0 (± 0.1)
Structure-SELFIES						
MAE	15 (± 0.4)	1.12 (± 0.04)	0.50 (± 0.03)	0.27 (± 0.02)	7.4 (± 0.3)	0.426 (± 0.008)
MAD / MAE	5.03 (± 0.1)	2.5 (± 0.09)	2.0 (± 0.1)	6.3 (± 0.4)	3.6 (± 0.1)	2.92 (± 0.05)

9. The authors have used PPAR γ protein structure with PDB ID 3G9E for this study. What is the reasoning behind using this structure instead of other structures available with more sequence coverage and/or better resolution? Why haven't authors opted to model the complete structure of protein?

>>> Thank you for your comments and questions regarding our choice of the PPAR γ protein structure including Aleglitazar as ligand (PDB ID 3G9E) for our study. We appreciate the opportunity to provide clarity on our selection and rationale for using this specific structure.

Aleglitazar as the Focus:

We chose PPAR γ with PDB ID 3G9E as the structural basis for our study due to our specific interest in Aleglitazar. Aleglitazar is among the most advanced PPAR agonists in clinical development, demonstrating validated antidiabetic and lipid-modulating efficacy in humans. Its proven in vivo activity, particularly in rodent models with successful translation towards Phase 2 trials in humans, makes it a compelling subject of investigation.

Proven In Vivo Activity:

The in vivo activity of Aleglitazar, particularly its modulation of glucose and lipid levels, provides a solid foundation for our study. The clinical relevance of Aleglitazar and its potential therapeutic applications motivated us to focus on understanding its molecular interactions at the structural level.

Structurally Conserved X-ray Structure:

The PPAR γ structure with PDB ID 3G9E exhibits a conserved and well-characterized X-ray structure. In our assessment, additional X-ray structures would not have significantly contributed to our understanding of Aleglitazar's interactions with PPAR γ . The existing structure provides sufficient resolution for our investigative purposes.

In summary, the choice of the PPAR γ structure with PDB ID 3G9E was driven by the clinical development and relevance of Aleglitazar, its advanced clinical status, proven in vivo activity, and the conserved nature of the available X-ray structure. We believe that this choice provides a robust foundation for our investigation into the molecular mechanisms underlying the therapeutic effects of Aleglitazar.

The following text is added to the main manuscript highlighting our choice and the relevance of PPAR γ :

“Dragonfly was utilized in a prospective manner for structure-based ligand design targeting human PPAR γ (PPAR γ , PDB-ID 3G9E [42]). The nuclear hormone receptor PPAR γ is one of the three peroxisome proliferator-activated receptors (i.e., PPAR $\gamma/\alpha/\delta$), that have been exploited as drug targets for combating multiple diseases, in particular metabolic syndrome-related disorders and cancer [43–45]. Activation of the PPARs by natural ligands or by synthetic PPAR agonists triggers the formation of heterodimers with members of the retinoid X receptor (RXR) family [46]. Upon recruitment of specific

cofactors, these heterodimers transactivate PPAR-responsive elements (PPREs) of target genes involved in insulin signaling, lipid and glucose metabolism, immune response, as well as cell cycle and differentiation [47, 48]. Several activators with different selectivity for the respective PPAR subtypes have reached advanced clinical trials or were introduced to the market.”

10. Why molecular dynamics simulation was not employed to validate more designs and then select top 2 designs? This could have helped find better compounds. The scores of the top 10 or 20 compounds should be mentioned in the manuscript.

>>> We thank the reviewer for the suggestion to employ molecular dynamics (MD) simulations for validation of the top design.

As described in **Reviewer 2 point 5** absolute protein–ligand binding free-energy perturbation (ABFEP) calculations using Schrödingers Free Energy Perturbation (FEP+) method were applied to the top 2 molecules revealing additional insights into protein-ligand binding of the novel compounds. The results are in the range of other known PPAR γ ligands with sub-micromolar target activity further advocating the relevance of the proposed molecules for PPAR γ modulation.

Since ABFEP come with high computational cost, applying this method to a large amount of molecules becomes infeasible. Therefore ABFEP has not been applied as scoring criteria of the de novo designed molecules.

11. Large amount of relevant data that should have been part of manuscript has been pushed to the supplementary information. The Methods section, especially, needs more information. More details of biological characterization and co-crystallization has to be added to the manuscript.

>>> We thank the reviewer for highlighting the importance of the biological characterization and co-crystallization of the presented study. Detailed description of the following methods has been added to the main part of the manuscript (Methods section):

“

Biological Characterization

Compounds 1 – 3 were characterized in a hybrid reporter gene assay for their agonistic effect on human nuclear receptors PPAR α / γ / δ , RXR α , FXR α , RAR α in HEK293T cells. Compound 1 was tested in an isothermal titration calorimetry (ITC) assay to measure direct binding affinity to the ligand-binding domain of PPAR γ . ADME properties were measured in standardized assays at Roche.

Hybrid reporter gene assays. PPAR activation was determined in uniform Gal4-hybrid reporter gene assays for the PPAR α , PPAR γ and PPAR δ isoforms in HEK293T cells (German Collection of Microorganisms and Cell Culture GmbH, DSMZ) which were transiently transfected with pFR-Luc (Stratagene, La Jolla, CA, USA; reporter) and pRL-SV40 (Promega, Madison, WI, USA; internal control) and one pFA-CMVhPPAR-LBD [99] clone, coding for the hinge region and ligand binding domain of the canonical isoform of human PPAR α , PPAR γ , PPAR δ or respectively. Cells were cultured in Dulbecco's modified Eagle's medium (DMEM), high glucose supplemented with 10% fetal calf serum (FCS), sodium pyruvate (1 mM), penicillin (100 U/mL), and streptomycin (100 μ g/mL) at 37 °C and 5 % CO₂ and seeded in 96-well plates (3 \times 10⁴ cells/well). After 24 h, medium was changed to Opti-MEM without supplements and cells were transiently transfected using Lipofectamine LTX reagent (Invitrogen) according to the manufacturer's protocol. Five hours after transfection, cells were incubated with the test compounds in Opti-MEM supplemented with penicillin (100 U/mL), streptomycin (100 μ g/mL) and 0.1% DMSO for 16 h before luciferase activity was measured using the Dual-Glo Luciferase Assay System (Promega) according to the manufacturer's protocol on a Tecan Spark luminometer (Tecan Deutschland GmbH, Germany). Firefly luminescence was divided by Renilla luminescence and multiplied by 1000 resulting in relative light units (RLU) to normalize for transfection efficiency and cell growth. Fold activation was obtained by dividing the mean RLU of a test compound by the mean RLU of the untreated control. All samples were tested in at least three biologically independent experiments in duplicates. For dose-response curve fitting and calculation of EC₅₀ values, the equation “[Agonist] versus response (variable slope - four parameters)” was used in GraphPad Prism (version 7.00, GraphPad Software, La Jolla, CA, USA) with fold activation data. The reference agonists GW7647 (PPAR α) [100, 101], pioglitazone (PPAR γ) [102, 103] and L165,041 (PPAR δ) [104, 105] were used to validate the assays and to monitor assay performance. Nuclear receptor selectivity profiling was

performed with corresponding pFA-CMV-hNR-LBD clones and suitable reference agonists on RAR α (pFA-CMV-hRAR α -LBD [106], 1 μ M tretinoin), LXR α (pFA-CMV-hLXR α -LBD [106], 1 μ M TO901317) and RXR α (pFA-CMV-h RXR α -LBD [107], 1 μ M Bexarotene).

Isothermal Titration Calorimetry (ITC). ITC experiments were conducted on an Affinity ITC instrument (TA Instruments, New Castle, DE) at 25 °C with a stirring rate of 75 rpm. PPAR γ LBD protein (30 μ M, prepared as described previously [108]) in buffer (20 mM Tris pH 7.5, 150 mM NaCl, 5% glycerol) containing 5% DMSO was titrated with the test compound (1) (100 μ M in the same buffer containing 5% DMSO) in 21 injections (1 \times 1 μ L and 20 \times 5 μ L) with an injection interval of 120 s. The test compound was titrated into buffer, and the buffer was titrated to the PPAR γ LBD proteins under otherwise identical conditions. The ITC results were analyzed using NanoAnalyze software (TA Instruments, New Castle, DE) with an independent binding model.

Protein-Ligand Co-Crystallization

The following construct was used for expression and co-crystallization: PPAR γ (L204-Y477) (UniProt ID: P37231-2): MGSS-6His-SG-TEV-(L204-Y477). Molecular weight: 33465 Da. Large-scale expression of human PPAR γ was conducted in *E. coli* BL-21 (DE3) cells (S110). Subsequently, co-crystals of PPAR γ were grown using 6 mg/mL protein in buffer: 20 mM Tris-HCl pH 8.0, 1 mM TCEP, 0.5 mM EDTA, and 1 mM design 1 mixed with equal amounts of reservoir: 0.1 M Tris-HCl pH 7.5 and 1.6 M ammonium sulfate (Figure S14). The structure was determined and refined, yielding the elucidated co-crystal structure with a resolution of 1.85 Å, as depicted in Figure 5 (Table S13 and Figure S15).

Off-Target Screening

To test the specificity of compounds 1 and 2, both were subjected to a panel screen against 50 safety-relevant off-targets [109]. Both compounds have shown a clear profile not reaching \geq 50% inhibition or binding at a concentration of 10 μ M, with the exception for PPAR γ (Table S9-S12).

”

12. The authors have not evaluated the stability of synthesized compounds. The authors may use liquid chromatography-mass spectrometry to check their stability. Determining the stability in blood plasma in vitro can be helpful to get an estimate of bioavailability of the compounds.

>>> We thank the reviewer for the suggestion to determine the stability of the de novo designed molecule, compound **1** and **2**. To evaluate the in vitro stability and clearance of the two designs, compound **1** and **2** underwent two additional in vitro assays.

1. CYP Dose Response Curves: Binding assay with seven pivotal cytochrome P450 isoenzymes (CYP, i.e., Cyp3A4, Cyp1A2, Cyp2B6, Cyp2C9, Cyp2D6, Cyp2C19, and Cyp2C8) were conducted. Compounds **1** and **2** demonstrated no interaction with the investigated CYPs up to concentrations of 20 μ M indicating high metabolic stability.

2. Hepatocyte Clearance: The hepatocytic clearance rate was determined at 19 μ L/min/ 10^6 cells for both compounds **1** and **2**. The determined hepatocytic clearance rates indicate high metabolic stability of both compounds as well as a desired pharmacokinetic profile.

Additional data provided by the two in vitro assays demonstrates the desired pharmacokinetic stability and in supports further clinical development of compound

Furthermore, the two designs have been profiled for their microsomal clearance in achieving high oral bioavailability in human, rat, and mouse microsomes. Thereby booth molecules enable the desired pharmacokinetic profile in both human and rodent efficacy studies.

The following text has been added to the **main part of the manuscript (results)** discussing all newly added ADMET data:

“Furthermore, the clearance values of compounds **1** and **2** in human, rat, and mouse microsomes were consistently low (≤ 10 μ L/min/mg protein) when compared to aleglitazar, suggesting the potential for achieving high oral bioavailability in both humans and rodents for efficacy studies. Compound clearance rates in human hepatocytes were determined at 19 μ L/min/ 10^6 cells for both compounds **1** and **2** (Table S14). Both metabolic and hepatocyte clearance suggest a sufficient metabolic profile, paving the way for further in vivo pharmacokinetic studies. Compounds **1** and **2** exhibited no interaction with the seven pivotal cytochrome P450 isoenzymes (CYP) - Cyp3A4, Cyp1A2, Cyp2B6, Cyp2C9, Cyp2D6, Cyp2C19, and Cyp2C8 - at dose-response experiments up to 20 μ M (Table S15).“

The following text, tables and figures has been added to the **Supporting Information**:

“... Furthermore, the evaluation of inhibition potential against cytochrome P450 isoenzymes (CYP) proteins was investigated. For seven selected CYP proteins (i.e., Cyp3A4, Cyp1A2, Cyp2B6, Cyp2C9, Cyp2D6, Cyp2C19, and Cyp2C8), the half maximal

inhibitory concentration (IC₅₀) was determined [43] (Table S15). The two designs (1 and 2) did not exhibit any relevant activity in the CYP assays.

”

Additionally, microsomal and hepatic clearance values have been added to Table S18.

Table S15: Half maximal inhibitory concentration (IC₅₀) in μ M for compounds **1** and **2** for seven selected CYP proteins, *i.e.*, Cyp3A4, Cyp1A2, Cyp2B6, Cyp2C9, Cyp2D6, Cyp2C19, and Cyp2C8.

Molecule	Enzyme	Substrate	IC ₅₀ (μ M)
1	CYP3A4	MIDAZOLAM	>20
	CYP1A2	PHENACETIN	>20
	CYP2B6	BUPROPION	>20
	CYP2D6	DEXTROMETHORPHAN	>20
	CYP2C9	DICLOFENAC	>20
	CYP2C19	S-MEPHENYTOIN	>20
	CYP2C8	AMODIAQUINE	>20
2	CYP3A4	MIDAZOLAM	>20
	CYP1A2	PHENACETIN	>20
	CYP2B6	BUPROPION	>20
	CYP2D6	DEXTROMETHORPHAN	>20
	CYP2C9	DICLOFENAC	>20
	CYP2C19	S-MEPHENYTOIN	>20
	CYP2C8	AMODIAQUINE	1.75

13. The authors have not performed cytotoxicity assays like MTT or WST to determine the cellular toxicity of the synthesized compounds, which is important to know before using them for therapeutic application. Was toxicity of compounds kept into consideration while designing them?

>>> We thank the reviewer for the suggestion to enhance the characterization of the de novo designs by examining their safety and cytotoxicity profiles. In response, Design **1** and **2** underwent additional ADMET (Absorption, Distribution, Metabolism, Excretion, and Toxicity) assays enriching our understanding of the compounds' clinical developability, safety and pharmacokinetic properties. The additional data advocates and supports the potential for further clinical development of compound **1** and **2**. The following additional assays were conducted and added to the main part of the manuscript as well as the SI.

- **PgP Transport Efflux:** The P-glycoprotein (Pgp) efflux ratio for compounds **1** and **2** have been determined, revealing ratios of 1.6 (15 nm/sec) and 1.2 (60 nm/sec), respectively. The observed efflux ratio indicates a desired pharmacokinetic behavior allowing for the application to multiple targeted cell and tissue types.
- **Plasma Protein Binding:** The unbound fractions for compounds **1** and **2** were determined at 0.42% and 0.21%. Such low values are attributed to the negatively charged carboxylic acid and similar to other carboxylic acid containing drug-like molecules.
- **Cytotoxicity:** Compounds **1** and **2** were tested for cytotoxicity on HEK293T cells at different time points as well as a broad array of cell and compound concentrations. No cytotoxic effects could be observed.

Furthermore, the two Designs **1** and **2** have been tested against binding for binding against 50 safety-critical off-targets. No binding could be observed for any of the investigated targets additionally supporting the desired safety profile of the two compounds.

The following text has been added to the **main part of the manuscript (results)** discussing the newly added ADMET data:

“ In terms of permeability through membranes, both molecules displayed favorable results in the parallel artificial membrane permeability assay (PAMPA), with permeation coefficients (PAMPAEFF) of $3.9 \text{ cm/s} \cdot 10^{-6}$ for compound **1** and $14 \text{ cm/s} \cdot 10^{-6}$ for compound **2**. Achieving sufficient cell permeability is crucial for targeting the PPAR γ receptor, located within the cell nucleus. Cellular permeability was confirmed in the P-glycoprotein (Pgp) efflux assay for compounds **1** and **2**, revealing values of 1.6 ($15 \text{ nm} \cdot \text{sec}^{-1}$) and 1.2 ($60 \text{ nm} \cdot \text{sec}^{-1}$), respectively. The observed efflux ratio indicates that compounds **1** and **2** are only interacting weakly with the Pgp transporter and thus, hold high potential to reach multiple cell and tissue types following a systemic application. Moreover, the unbound fractions of compounds **1** and **2** were determined at 0.42% and

0.21%. Such low unbound fractions are attributed to the negatively charged carboxylic acid, similar to other drug-like molecules containing carboxylic acid groups [42].

“

and

“Moreover, both compounds presented a favorable profile in an expansive panel screen assessing multiple safety-critical off-targets. Importantly, none of the targets exhibited binding or inhibition above 60% at a compound concentration of 10 μM (Table S13 - S16). Furthermore, compounds 1 and 2 did not indicate any cytotoxicity on HEK293T cells at different time points as well as a broad range of cell numbers and compound concentrations (Figure 5e, Figure S16). Collectively, the computer-designed compounds 1 and 2 showcase a promising drug-like profile, signifying substantial potential for advancement in further drug development.

“

Additionally panel e in Figure 5 was added illustrating the cytotoxicity results:

“

e: Cytotoxicity of compounds 1 and 2 on HEK293T cells for two time points (i.e., 16h and 24h), 10 different concentrations (i.e., 0.05 - 20 μM) and 10000 cells / well (N = 3). Scale reference: The axes are scaled through neutral control (i.e., Dimethylsulfoxid [DMSO], set to 0) and inhibitor control wells (i.e., 20 μM Staurosporine [54], set to -100).

”

The following text has been added to the **main part of the manuscript (Methods)**:

“

Cytotoxicity Assay on HEK293T Cells

HEK293T cells were seeded at the indicated number per well in DMEM-high glucose, complemented with glutamax, pen/strep, and 10% FBS, in a total of 40 μ l of medium. The cells were incubated overnight at 37°C. Compounds were added to the cells at the indicated concentrations, resulting in a final Dimethylsulfoxid (DMSO) concentration of 0.2%. The compounds were incubated on the cells for either 16 hours or 24 hours. At the specified time point, the medium was carefully removed from the vessel, leaving only 25 μ l in the wells. Celltiter-glo (CTG) reagent (G7572, Promega) was prepared according to the manufacturer's instructions. Plates with cells were equilibrated at room temperature for 30 minutes. Subsequently, 25 μ l of CTG reagent was added to the cells. The plates were then shaken for 2 minutes and incubated for an additional 15 minutes at room temperature. Luminescence was read afterward with BG Pherastar.

“

The following text, tables and figures has been added to the **Supporting Information**:

“

SI 11 Absorption Distribution Metabolism and Excretion Data

To assess the drug-like characteristics of the newly designed PPAR modulators, a series of tests were conducted covering various aspects of absorption, distribution, metabolism, and excretion (ADME) properties. Specifically, these assessments included the parallel artificial membrane permeability assay (PAMPA) with permeation coefficients (PAMPAP_{EFF}) [38], LogD measurements, microsomal clearance [39, 40], hepatocyte clearance [41], P-glycoprotein (Pgp) efflux ratio and binding [42], and the unbound free fraction [42]. Table S14 provides an overview of the observed ADME values for compounds 1 and 2.

Table S18: Absorption distribution metabolism and excretion (ADME) properties of compounds **1** and **2**. The values for PAMPAP_{EFF}, LogD, protein binding, P-glycoprotein (Pgp) efflux ratio and binding, and hepatocyte clearance are presented as the mean and standard deviation, with a sample size of N=3, *i.e.*, 3 technical replicates each. The values for microsomal clearance and CYP-inhibition represent single point measurements.

	Aleglitazar	1	2
Molecular weight (g/mol)	437.5	434.5	432.5
Lipophilicity (LogD pH 7.4)	1.4 (\pm 0.3)	1.5 (\pm 0.3)	1.7 (\pm 0.6)
PAMPAP _{EFF} (cm/s*10 ⁻⁶)	6.0 (\pm 1.2)	14 (\pm 3)	3.9 (\pm 0.4)
Microsomal clearance, human (μ L/min/mg protein)	\geq 10	\geq 10	\geq 10
Microsomal clearance, mouse (μ L/min/mg protein)	\geq 10	\geq 10	\geq 10
Microsomal clearance, rat (μ L/min/mg protein)	\geq 10	\geq 10	\geq 10
Protein binding human (% free fraction) (%)	-	0.21	0.42
PgP Transport mouse apical efflux ratio	-	1.6	1.2
PgP Transport mouse permeability (nm/sec)	-	15	60
Hepatocyte clearance human (μ L/min/10 ⁶ Cells)	-	19	19

Compounds 1 and 2 were validated for cytotoxicity on HEK293T cells. Figure S16 depicts the measured cytotoxicity at two time points (i.e., 16h and 24h), 10 different concentrations (i.e., 0.05 - 20 μM), and 3 different numbers of cells per well (i.e., 3000, 5000, and 10000).

Figure S16: Cytotoxicity assay (N = 3) on HEK293T cells for compounds 1 and 2 for two time points (i.e., 16h and 24h), 10 different concentrations (i.e., 0.05 - 20 μM), and 3000 - 10000 cells per well. Scale reference: The axes are scaled through neutral control (i.e., Dimethylsulfoxide [DMSO], set to 0) and inhibitor control wells (i.e., 20 μM Staurosporine [44], set to -100).

“

14. Currently there's too much technical jargon for an average reader. Wherever possible, manuscript should be simplified and/or explained for better readability.

>>> We thank the reviewer for this remark. The technical jargon of the main part of the manuscript as well as the Supporting Information has been reduced to a minimum. The current vocabulary should satisfy the readability of a broad audience.

My decision: Major revision

Reviewer #3 (Remarks to the Author):

The authors propose a computational approach based on interaction deep learning for ligand-and structure-based drug molecule generation. This approach takes advantage of the unique strengths of graph neural networks and chemical language models to provide an alternative for application-specific reinforcement, transfer, or few shot learning.

>>> Thank you for your kind review and your most helpful constructive criticism. Peer-review at its best! We hope that by addressing your remarks, we restassured your confidence that our revised manuscript can be accepted.

1. The authors should disclose the model's hyperparameters and training process in the supplementary information to the paper so that researchers can better reproduce the results of the paper.

>>> We thank the reviewer for his valuable suggestion regarding the disclosure of the model's hyperparameters and training process. Ensuring reproducibility is a priority for us, and we appreciate your emphasis on the importance of providing detailed methodological information.

In response to your comment, we have added comprehensive details about the hyperparameters and the neural network training setup to the main part of the manuscript.

“Hyperparameters

The selected hyperparameters for the neural network led to a combined count of trainable parameters amounting to 6.94 million (3.49 million for the GTNN encoder and 3.45 million for the LSTM decoder) for the ligand-based design Dragonfly model. Similarly, the structure-based design Dragonfly model encompassed 7.01 million trainable parameters (3.56 million for the GTNN encoder and 3.45 million for the LSTM decoder).”

Furthermore, we have elaborated on the neural network training process in SI1, detailing the software and hardware used, the batch sizes for different design approaches, the optimizer and its learning rate, the loss function, the decay factor, and the smoothing factor applied during training.

“SI1 Neural Network Training

Neural network training was conducted using PyTorch Geometric (version 2.0.2)[1] and PyTorch (version 1.10.1+cu102)[2]. The training process occurred on a GPU, specifically the NVIDIA Tesla V100-SXM2 32 GB, for a total duration of 120 hours. A batch size of 128 was used for ligand-based design, while a batch size of 48 was employed for structure-based design. The optimization of the neural networks was performed using the Adam stochastic gradient descent optimizer [3], with a learning rate set at 0.0005. Cross-entropy loss was utilized as the loss function, and a decay factor of 0.5 was applied after every 10 epochs. Additionally, an exponential smoothing factor of 0.7 was incorporated into the training process. All the models used in this study were trained on the Euler computing cluster at ETH Zurich, Switzerland. “

2. Why was PPAR ligand selected for validation? The author should explain this in the paper.

>>> We thank the reviewer for their question regarding our selection of the PPAR γ ligand for validation in our study. The choice was made with careful consideration of both the clinical significance and the structural characteristics of the ligand-receptor complex.

Aleglitazar was selected as the focus of our study due to its status as one of the most advanced PPAR agonists in clinical development. Its antidiabetic and lipid-modulating efficacy has been validated in human clinical trials, with promising results from Phase 2 studies. This clinical relevance was a primary factor in our decision, as we aimed to investigate a compound with potential therapeutic applications.

The *in vivo* activity of Aleglitazar, particularly its effects on glucose and lipid metabolism, has been well-documented. This provided a strong rationale for examining its molecular interactions with PPAR γ in detail. By understanding these interactions, we aim to elucidate the molecular basis for its observed biological effects, which could inform the development of new therapeutic strategies. Furthermore, the PPAR γ structure with PDB ID 3G9E was chosen because it represents a conserved and well-characterized X-ray structure of the receptor in complex with Aleglitazar. We determined that additional X-ray structures would not offer a significant advantage for our specific research objectives. The chosen structure provides the necessary resolution to investigate the binding interactions of Aleglitazar with PPAR γ , which is crucial for our study.

In conclusion, the selection of PPAR γ with Aleglitazar (PDB ID 3G9E) for our study was based on the clinical importance of Aleglitazar, its proven *in vivo* activity, and the suitability of the available X-ray structure for our research aims. We believe that this selection offers a solid foundation for our investigation into the molecular mechanisms of Aleglitazar's therapeutic effects.

The following text is added to the main manuscript highlighting our choice and the relevance of PPAR γ :

“Dragonfly was utilized in a prospective manner for structure-based ligand design targeting human PPAR γ (PPAR γ , PDB-ID 3G9E [42]). The nuclear hormone receptor PPAR γ is one of the three peroxisome proliferator-activated receptors (i.e., PPAR γ / α / δ), that have been exploited as drug targets for combating multiple diseases, in particular metabolic syndrome-related disorders and cancer [43–45]. Activation of the PPARs by natural ligands or by synthetic PPAR agonists triggers the formation of heterodimers with members of the retinoid X receptor (RXR) family [46]. Upon recruitment of specific cofactors, these heterodimers transactivate PPAR-responsive elements (PPREs) of target genes involved in insulin signaling, lipid and glucose metabolism, immune response, as well as cell cycle and differentiation [47, 48]. Several activators with different selectivity for the respective PPAR subtypes have reached advanced clinical trials or were introduced to the market.”

3. In the Methods section, the author simply introduces some methods used in DRAGONFLY. I think the author should introduce the design idea of the model in detail, so that researchers can have a deeper understanding of the model. Rather than simply introducing some methods used.

>>> We are grateful to the reviewer for the suggestion to provide a more detailed introduction to the design idea of the Dragonfly model in the Methods section. We recognize the importance of enabling researchers to gain a deeper understanding of the model's conceptual framework and operational mechanics.

In response to this feedback, we have expanded the Methods section to include comprehensive descriptions of the neural network architecture and dataset preprocessing, as well as a more thorough explanation of the training process. These enhancements are intended to provide clarity on the design principles and implementation details of Dragonfly .

Neural network architecture

The detailed model architecture and the hyperparameter are described in the Methods section of the main manuscript.

This information is now comprehensively detailed in the following subsections: "Neural Network Architecture," "Graph Transformer Neural Network," "Long-Short-Term Memory Neural Network," "Molecule Sampling," "Atom Featurization," and "Hyperparameters."

Dataset preprocessing

Key to the introduced model is data set preprocessing and the generation of the drug target interactome from the two databases, i.e. ChEMBL and PDBBind. This information is now comprehensively detailed in the following subsections: "Drug-Target Interactome Preprocessing", "Preprocessing ChEMBL Data", "Preprocessing PDBbind"

Training process

We have elaborated on the neural network training process in (SI1 Neural Network Training) , detailing the software and hardware used, the batch sizes for different design approaches, the optimizer and its learning rate, the loss function, the decay factor, and the smoothing factor applied during training.

Moreover, we would like to highlight that the code for the Dragonfly application and training will be made available upon acceptance of the manuscript, allowing other researchers to replicate and build upon our work.

We hope that these additions and clarifications will satisfy the reviewer's request and contribute to a more comprehensive understanding of the Dragonfly model.

4. The authors should compare the performance with other drug design models to highlight the superiority of the models.

>>> We thank the reviewer for this constructive comment. In response to the request for a comparison with other drug design models, we acknowledge the importance of benchmarking our Dragonfly model against existing approaches to underscore its performance and potential advantages.

Dragonfly represents a novel methodology in the realm of de novo drug design, integrating both structure- and ligand-based strategies within a single model framework without the need for transfer learning. This integration, coupled with the model's ability to directly incorporate physical and chemical property constraints sets Dragonfly apart from existing methods. Given the unique combination of features in Dragonfly, direct comparisons with other models that do not share this integrative approach are challenging.

Nevertheless, to address the reviewer's point, we have conducted a thorough comparison with a state-of-the-art fine-tuned recurrent neural network (RNN) model, which is a representative of advanced generative models in drug design that typically require transfer learning and target-specific fine-tuning. The comparison is detailed in Table 1 of the manuscript, where we present a quantitative evaluation of both models across several criteria relevant to the drug selection process.

The results in **Table 1** reveal that Dragonfly not only matches but in some cases surpasses the performance of the fine-tuned RNN model. Specifically, Dragonfly demonstrates a higher percentage of generated molecules meeting the predefined selection criteria, which suggests that our model's generative capabilities are inherently strong.

We believe that the inclusion of these comparative results will satisfactorily address the reviewer's concerns and highlight the innovative aspects of our approach within the context of current drug design methodologies.

REVIEWERS' COMMENTS

Reviewer #1 (Remarks to the Author):

The authors have improved the quality of the manuscript by the additional studies. There are very few aspects left, which should be addressed.

Fig.1b, brown instead of red circle.

I recommend to add more technical details for the compound generation with RNN in the SI material.

Reviewer #2 (Remarks to the Author):

Authors have satisfactory answered all the queries and I am confident with all the revisions. This paper may be accepted in the current form.

Reviewer #3 (Remarks to the Author):

The revision is satisfactory, the manuscript is acceptable in its current form

We are grateful to all three reviewers for their exceptional peer-review efforts, which have significantly enhanced the quality of our study.

Reviewer #1 (Remarks to the Author):

The authors have improved the quality of the manuscript by the additional studies. There are very few aspects left, which should be addressed.

Fig.1b, brown instead of red circle.

I recommend to add more technical details for the compound generation with RNN in the SI material.

We thank reviewer 1 for the final remarks and for critically conducting a detailed review of the caption in Figure 1 and the SI.

The term "red" in the caption of Figure 1 has been replaced with "brown."

Furthermore, the SI now includes additional information elucidating the detailed setup utilized for compound generation with the fine-tuned RNN. The following text has been added:

"For the RNN baseline, a model pre-trained on ChEMBL24 [26] was fine-tuned using individual templates with standard settings. These settings comprised fine-tuning for up to 40 epochs, a learning rate of 10^{-4} , and compound generation with a temperature of $T = 0.7$. Four different numbers of fine-tuning epochs were compared: 10, 20, 30, and 40. Notably, 30 fine-tuning epochs yielded the best trade-off between molecules with desired novelty and desired properties. Consequently, 30 fine-tuning epochs were chosen for the subsequent comparison."

Reviewer #2 (Remarks to the Author):

Authors have satisfactorily answered all the queries and I am confident with all the revisions. This paper may be accepted in the current form.

Thank you for your positive feedback and support for our paper's acceptance.

Reviewer #3 (Remarks to the Author):

The revision is satisfactory, the manuscript is acceptable in its current form

We appreciate your approval of the revisions and acceptance of our manuscript in its current form.